# The Role of APOBECs in Viral Replication

**DOI:** 10.3390/microorganisms8121899

**Published:** 2020-11-30

**Authors:** Wendy Kaichun Xu, Hyewon Byun, Jaquelin P. Dudley

**Affiliations:** 1Department of Molecular Biosciences, The University of Texas at Austin, Austin, TX 78712, USA; kx475@utexas.edu (W.K.X.); hbyun@utexas.edu (H.B.); 2Interdisciplinary Life Sciences Graduate Program, The University of Texas at Austin, Austin, TX 78712, USA; 3LaMontagne Center for Infectious Disease, The University of Texas at Austin, Austin, TX 78712, USA

**Keywords:** APOBEC, RNA virus replication, DNA virus replication, AID, HIV-1 Vif, genome hypermutation, MMTV Rem, G-to-A mutations, viral variants

## Abstract

Apolipoprotein B mRNA-editing enzyme catalytic polypeptide-like (APOBEC) proteins are a diverse and evolutionarily conserved family of cytidine deaminases that provide a variety of functions from tissue-specific gene expression and immunoglobulin diversity to control of viruses and retrotransposons. APOBEC family expansion has been documented among mammalian species, suggesting a powerful selection for their activity. Enzymes with a duplicated zinc-binding domain often have catalytically active and inactive domains, yet both have antiviral function. Although APOBEC antiviral function was discovered through hypermutation of HIV-1 genomes lacking an active Vif protein, much evidence indicates that APOBECs also inhibit virus replication through mechanisms other than mutagenesis. Multiple steps of the viral replication cycle may be affected, although nucleic acid replication is a primary target. Packaging of APOBECs into virions was first noted with HIV-1, yet is not a prerequisite for viral inhibition. APOBEC antagonism may occur in viral producer and recipient cells. Signatures of APOBEC activity include G-to-A and C-to-T mutations in a particular sequence context. The importance of APOBEC activity for viral inhibition is reflected in the identification of numerous viral factors, including HIV-1 Vif, which are dedicated to antagonism of these deaminases. Such viral antagonists often are only partially successful, leading to APOBEC selection for viral variants that enhance replication or avoid immune elimination.

## 1. Introduction

Apolipoprotein B mRNA-editing enzyme catalytic polypeptide-like (APOBEC) proteins constitute a protein family that deaminates cytosines within nucleic acids. This review begins with a description of APOBEC proteins and their enzymatic activity, structure, and function. This description is followed by a discussion of the primarily inhibitory relationship between APOBECs and virus replication. This inhibitory relationship is exemplified by the multiple mechanisms used by viruses to thwart APOBEC function as well as the induction of APOBEC proteins by type I interferons (IFNs), which induce the antiviral state in infected and neighboring cells. Due to the large numbers of papers on APOBEC enzymes, we will focus on publications since our previous review with the Harris laboratory [1].

### 1.1. APOBEC Enzyme Activity

Humans and non-human primates have 11 members of the *APOBEC* gene family, namely *APOBEC1* (*A1*), *APOBEC2* (*A2*), *APOBEC3A* (*A3A*), *APOBEC3B* (*A3B*), *APOBEC3C* (*A3C*), *APOBEC3D* (*A3D*), *APOBEC3F* (*A3F*), *APOBEC3G* (*A3G*), *APOBEC3H* (*A3H*), *APOBEC4* (*A4*) and activation-induced cytidine deaminase (*AICDA*). Other animal species have fewer APOBEC genes [1,2]. All species from fish to mammals encode the primordial APOBEC member, activation-induced cytidine deaminase (AID), but *A2* also is an ancestral *APOBEC* gene [3]. The *A1* gene likely evolved by duplication of the *AICDA* gene and is tightly linked on the same chromosome [3]. The APOBEC proteins (A1, A3, and AID) have deamination activity on cytosines within polynucleotides [3]. A4 appears to lack deaminase activity [2], whereas the enzymatic activity of A2 is controversial [4]. Active APOBECs induce deamination of cytosine to uracil within RNA or DNA targets by a zinc-dependent hydrolytic mechanism using a glutamic acid within the conserved HXE motif [2,3]. A zinc-stabilized hydroxide ion (-OH) group from water attacks the NH_2_ at the 4-position of the cytosine aromatic ring to form uracil with the release of NH_3_ (Figure 1A) [5,6,7,8,9]. This basic mechanism is preserved from bacterial cytosine deaminases to mammalian enzymes [3].

As the name implies, *A1* was the first identified member of the *APOBEC* gene family. The *A1* gene is present in mammals and evolved after the appearance of *AID* and *A2* [3]. The A1 protein is responsible for the RNA editing of a specific cytosine (C6666) to uracil within apolipoprotein B (*APOB*) mRNA [11], leading to a stop codon and truncation of the APOB-100 protein to produce APOB-48 [12]. APOB-100 synthesized in the liver is the primary protein component of low-density lipoproteins in plasma, whereas mRNA editing and APOB-48 production in the small intestine is required for chylomicron secretion [13]. Thus, *APOB* mRNA editing affects the risk of atherosclerosis [14,15]. In vitro, A1 binds RNAs with a preference for AU-rich sequences, whereas much greater deamination specificity is observed in tissues. This observation has been explained by the discovery that A1 deaminase activity in vivo requires an RNA-binding component. A1 must associate with an RNA-binding protein cofactor, either A1 complementation factor (A1CF) [16,17] or RNA-binding motif protein 47 (RBM47) [18], for specific substrate targeting. Cofactor binding to RNA requires both a 5′ element and a 3′ 11-nucleotide mooring sequence relative to the targeted cytidine in *APOB* mRNA [17]. The mooring motif is essential for the C-to-U site-specific editing of the *APOB* transcript [19] in AU-rich RNA [20]. C-to-U editing by A1 is a nuclear event [13] and may target the 3′ untranslated regions (UTRs) of many transcripts [15]. Transcriptomic analysis showed that A1 prefers to edit a cytidine flanked by adenine nucleotides in the +1 and −1 positions for RNA editing [14,15]. A1 editing or simply binding to the 3′ UTRs of multiple cellular mRNAs has the potential for antagonism of viruses.

Other catalytically active APOBEC proteins (A3s and AID) primarily deaminate single-stranded DNA (ssDNA). Unlike its activity on RNA substrates, A1 can perform C-to-U editing on ssDNA without a cofactor [21,22,23]. A3 and AID require at least five contiguous deoxynucleotides for deamination, in which the target deoxycytidine is preceded by three deoxynucleotides at the 5′ end and one deoxynucleotide at the 3′ end [24,25]. APOBEC proteins have different preferences for cytidine context within the ssDNA. For example, A3G prefers that the targeted cytidine is preceded by another cytidine (5′ CC 3′), whereas deamination by the remaining A3 proteins (A3A/B/C/D/F/H) occurs primarily in the 5′ TC 3′ context. Another APOBEC protein, AID, prefers a pyrimidine and purine prior to the target cytidine (5′ WRC 3′) [26,27,28,29]. These characteristics have been used to predict the enzymes that have targeted viral genomes for deamination.

### 1.2. APOBEC Protein Structures

#### 1.2.1. Nucleotide-Binding Activity

The structures of multiple APOBEC enzymes have been determined, although no crystal structures are currently available for A1 binding to RNA. Structural studies of A3A or A3B bound to ssDNA have revealed that targeting differences are likely due to substrate DNA binding in a U-shaped conformation. Binding to DNA leads to the −1 base being flipped out of the catalytic active site, and the targeted cytosine is inserted deep into the zinc-binding active site pocket [30,31]. The −1 base forms a specific hydrogen bond with the protein, which explains enzyme targeting preferences [31]. In addition to nucleotide preferences, the −2 and +1 bases also affect deamination efficiency. However, details are not completely understood [1,2,25,32,33,34] (see following sections).

A recent study has shown that AID preferentially binds to structured DNA-G quadruplexes (G4) during class switch recombination (CSR) to generate antibody diversification [35]. In these experiments, AID targeted cytidines at the third position in a 5′ overhang adjacent to the G4 using a bifurcated substrate-binding surface, thus capturing two adjacent single-stranded overhangs simultaneously. The G4 substrates also induced cooperative AID oligomerization, possibly to enhance deamination efficiency [35]. This binding mode may be unique among the APOBEC family since AID is the only identified member known to bind specific structured DNA [36].

APOBEC binding of non-substrate nucleic acids has been shown to regulate deamination activities in multiple ways. For example, A3F binds non-substrate ssDNA within a positively charged nucleic acid-binding site, which is distal to the active site within its C-terminal domain. Mutation of this binding site impaired its catalytic activity [37]. RNAs also bind APOBEC proteins to either inhibit or promote their substrate recognition and DNA deaminase activity. In in vitro experiments, A3G binding to RNA with at least 25 nucleotides displaced A3G binding to ssDNA and inhibited deaminase activity [38,39]. A3G binds a variety of RNA sequences to form ribonucleoprotein particles (RNPs), which inhibits deaminase function by sequestration into cellular P-bodies [38,39,40,41,42,43]. However, recent structural studies showed that purified A3H remained bound to an RNase-insensitive double-stranded RNA to maintain an RNA-mediated A3H dimer, and this supported A3H deaminase activity on DNA [44,45,46,47]. A3s bind some cellular RNAs, such as 7SL1 non-coding RNA (ncRNA), for efficient assembly into virions [48,49,50]. RNA-binding preferences for A3 enzymes may mimic the binding preference of viral nucleocapsid proteins. These preferences change after HIV-1 infection [51]. Overall, how binding of different RNAs affects APOBEC function remains unclear (see also a recent review from Salter et al. [36]).

#### 1.2.2. APOBEC Domain Structure

Each protein in the APOBEC protein family contains either one or two zinc-coordinating (Z) domains (Figure 1B) [1,3,52,53]. These domains have the consensus sequence HXE-X_25–30_-PCX_2–4_C in which the histidine and cysteine residues are involved in zinc coordination [54]. In the A3 subfamily, Z domains are categorized into three distinct phylogenetic groups (Z1, Z2, and Z3) [55]. A3A (Z1), A3C (Z2) and A3H (Z3) are composed of a single active cytidine deaminase Z domain, whereas A3B (Z2-Z1), A3D (Z2-Z2), A3F (Z2-Z2), and A3G (Z2-Z1) are all composed of an N-terminal inactive Z domain and a catalytically active C-terminal Z domain [56]. Although the double-domain proteins have only one catalytically active domain, both likely contribute to contacts with substrate DNA [54]. A3H has seven haplotypes with varying stabilities and enzymatic efficiencies [57]. AID and A1 have a single catalytically active zinc-coordinating domain [1,53].

All 11 primate APOBEC proteins share a conserved cytidine deaminase fold in their core catalytic active domain (Figure 1C) [1,52,53]. This conserved domain is composed of five mixed β sheets with either parallel or antiparallel hydrogen bonding. These β sheets are stabilized by six α-helices packed around either face, arranged sequentially as α1-β1-β2-α2-β3-α3-β4-α4-β5-α5-β6-α6 [1,53]. Loops that line the cytidine-binding pocket are the least conserved regions within this domain, which likely define the differences among APOBEC functions, such as substrate specificity and regulation [1,52,53]. Sequence alignments of all catalytically active APOBECs that function on ssDNA in humans (A3 proteins and AID) reveal that residues essential for catalysis are identical in all domains [52]. These residues are the zinc-coordinating histidine residue and the catalytic glutamate at the HXE motif with two cysteines at the N-terminal ends of α2 and α3 helices of the conserved secondary structure [1,52,53]. This type of organization is evolutionarily conserved with the yeast deaminase Cdd1, which acts on both RNA and ssDNA [54]. APOBEC loops that connect the conserved secondary structural elements vary in amino acid composition, length, and spatial conformation, especially in loops L1, L3, and L7 [53]. These three loops control the substrate selectivity among different APOBECs [36,53]. The L1 and L7 loops are spread apart to allow access of the target deoxycytidine base to the open catalytic site [58]. The L3 loops of A3G and A3A have been shown to coordinate a secondary zinc ion, which allosterically enhances the deaminase activity [59]. The zinc ion coordination through intermolecular oligomerization in adjacent L3 loops has been observed in the A3G structure [10] and at the A3A dimerization interface [60]. The L7 loop forms multiple hydrogen bonds with the Watson–Crick (WC) face of the base at the −1 position of the targeted cytidine [36]. A3G also forms additional water-mediated hydrogen bonds with the WC face of the −2 position, which explains the preference for deoxycytidine by A3G at the −2 position [61].

For dual domain-containing human APOBECs, such as A3G, the N-terminal domain is catalytically inactive, yet essential for deamination activity. In isolation, the C-terminal domain of A3G (CD2) has low or no deamination activity and requires the N-terminal domain for full activity [62]. Alterations of amino acid residues around the pseudo-catalytic site of the A3G N-terminal domain yield a mutant enzyme that displays reduced deamination rates compared with the wild-type A3G [63]. This N-terminal A3G cytidine deaminase domain (CD1) is required for binding to retroviral RNA and encapsidation into virions for inhibition of viral replication [62]. Examination of the crystal structure of rhesus macaque APOBEC3G (rA3G) revealed that CD1 allows A3G dimer formation to increase DNA binding [64]. The folded structure of CD1 is similar to the catalytically active domain (CD2), composed of six major α helices and five core β strands [64]. Both rA3G and human A3G (hA3G) CD1 domains have a positively charged surface on the opposite side from the C terminus, which allows binding of negatively charged nucleic acids [64]. Another recent study of full-length rA3G and recombinant rA3G-CD1-hA3G-CD2 also showed that A3G dimerization occurred through CD1-CD1 interaction [65]. Mutations of the positively charged surface and/or dimerization interface on CD1 reduced RNA and DNA association, resulting in loss of deaminase activity [65]. Although we still lack the complete mechanistic basis for requiring full-length A3G for deamination, the CD1 domain likely provides structural support for the catalytically active domain, especially for nucleic acid binding and dimerization/oligomerization [14,36]. More studies are needed to fully understand the role of CD1.

As noted above, the catalytic activity of A2 remains questionable. Deaminase assays with A2 revealed no DNA mutator activity in *E. coli* [22]. However, overexpression of A2 in transgenic mice suggested selective RNA editing [66]. Although initial X-ray crystal structures of an N-terminally truncated A2 protein indicated a V-shaped homotetramer [67], more recent studies using NMR showed that full-length A2 is a monomer in solution [4]. A4 also lacks deaminase activity. Overexpression of A4 in yeast and bacteria did not yield cytidine deaminase activity on DNA [68]. Nevertheless, the structures of both human and chicken A4 proteins, like other APOBECs, have a similar Zn-coordinating motif (HXE … C(X)_2–6_C), including five β strands and six α helices. The presence of a six amino-acid linker is specific for A4 [68]. Why some APOBECs (A2, A3A, and A3C) function as monomers, whereas others (A3B, A3D, A3F, A3G, A3H, and AID) require multimerization, remains under investigation [69]. Together, these studies suggest that fundamentally similar APOBEC structures can produce deaminase activity on either RNA or DNA targets. Self-association and interactions with cellular proteins and RNA regulate deaminase activity.

### 1.3. APOBEC Expression and Localization

APOBEC enzymes are known to be expressed in a tissue- and cell-specific manner. As mentioned previously, A1 editing of *APOB* mRNA occurs in the enterocytes of the small intestine to express the truncated APOB-48 protein, which lacks the C-terminal portion of the protein recognized by low density lipoprotein receptors [70]. A1 editing of *APOB* mRNA was first described in the small intestine, and many transcripts are edited in intestine and liver. Nevertheless, A1-induced changes also occur in macrophages and dendritic cells (DCs) [71], which often are the first cell types encountered by infecting viruses. Interestingly, mouse A1 (mA1) editing in DCs was altered in cells treated with lipopolysaccharide (LPS) [71], which stimulates TLR4 signaling and IFN production [72]. Editing preferentially occurred in the 3′ untranslated regions (UTRs), some at specific cytidines, but other transcripts were edited at multiple positions (hyperedited) [71]. Characterization of A1-knockout mice revealed that CD11c+ and CD11c- microglia in the central nervous system had differences in gene expression, notably decreased editing of the 3′ UTR of the *Lamp2* mRNA. Decreased mRNA editing yielded lower Lamp2 protein levels, which was accompanied by impaired lysosomal and innate immune function [73]. A1 primarily is localized to the cytoplasm [54], presumably the intracellular site of mRNA editing.

Compared to A1, the more ancient cytidine deaminase, A2, is relatively uncharacterized. A2 is expressed in a human liver cell line and is induced by tumor necrosis factor alpha (TNFα) [66]. Wild-type mice have the highest A2 levels in skeletal and cardiac muscle, whereas transgenic mice that overexpress A2 from the chicken β-actin promoter showed expression in most tissues. Transgenic animals developed hepatocellular and lung carcinomas [66]. The non-tumor liver tissues showed about a 5-fold increase in mutations distributed throughout *Eif4g2* and *Pten*, but not *Tp53* or *Bcl6* mRNAs. Although mutations were not observed in DNA, most RNA mutations did not affect cytidines [66], so the results remain difficult to explain. Studies in A2-knockout mice revealed defects in skeletal muscle, including enlarged mitochondria with defective function and increased mitophagy [74]. A2 expression in rat skeletal muscle appears to be cytosolic [75]. Therefore, A2 may edit specific transcripts that affect mitochondrial function in muscle, but potentially other tissues.

Many of the A3 enzymes are inducible. Interferon alpha (IFNα), a type I IFN, is produced by hematopoietic cells [76]. IFNα increases expression of A3A, A3G, and A3F in monocyte-derived dendritic cells (MDDCs) [77]. Another type I IFN, IFNβ, which is expressed in most cell types [76], induces A3A and A3G [78], whereas type I IFNs likely do not induce A3B and A3C [79]. IFNγ, a type II IFN that is primarily expressed by NK cells and activated T cells [80], increases A3G expression in monocytes and macrophages [81,82] and also increases A3A in the presence of TNFα [81,83]. A variety of external stimuli, including LPS, dsRNA, defensins, chemokines, and cytokines, result in elevated A3 levels [84]. Both DCs and macrophages are among the first immune cells that contact infectious viruses. Indeed, viral infection is well known to induce APOBEC expression, often through IFN induction [78,85]. Human A3D, A3F, A3G, and A3H are localized to the cytoplasm, although some A3H haplotypes are nuclear [86,87]. A3A shuttles between the nucleus and cytosol, but A3B is primarily nuclear [13]. Both A3A and A3B expression have been associated with genome hypermutations in human cancers [2].

Unlike humans, mice have a single A3 gene (mA3). However, mice have two different mA3 isoforms that are expressed in different strains [88,89,90,91]. In BALB/c mice, all nine exons encode a longer isoform of A3, whereas in C57BL/6 (B6) mice, a shorter isoform also is made due to exon 5 removal during splicing [88,89]. Both isoforms of mA3, like human A3G (hA3G), have N-terminal and C-terminal deaminase domains. In contrast to hA3G, mA3 has the catalytic domain within the N-terminus rather than the C-terminus [92]. mA3 is expressed in murine lymphocytes and mammary epithelial cells [93], which are cell types required for infection by the betaretrovirus Mouse Mammary Tumor Virus (MMTV) [93,94], but also murine testes [95]. The mA3 enzyme is localized to the cytoplasm where retroviral RNA packaging occurs [96].

A4 is a more evolutionarily recent APOBEC family protein that is expressed primarily in mammalian testis, suggesting that this enzyme may edit mRNAs involved in spermatogenesis [97]. In chickens, A4 is expressed in multiple immune cells, but also is induced by virus infection [98]. Therefore, an open question is whether the A4 enzyme, with a similar structural organization as other APOBECs, has catalytic and antiviral activity.

AID is another ancient member of the APOBEC family. AID transcription is inducible in naïve B cells after B-cell receptor (BCR), Toll-like receptor (TLR), and cytokine signaling through the NFκB pathway [99]. AID expression is further induced in dark and outer zone germinal center B cells and in large extrafollicular B cells through CD40 and BCR signaling [100]. In germinal center B cells, AID produces somatic hypermutation (SHM) of the immunoglobulin variable regions as well as class switch recombination (CSR). SHM and CSR are critical for the selection of high affinity antibodies with optimal effector functions in response to infections by multiple pathogens. Although AID is best known for its role in antibody affinity maturation, AID expression functions together with the recombination-activating gene 2 (RAG2) enzyme in bone marrow-derived immature B cells to eliminate self-reactive B cells [101]. In addition, several reports indicate that AID is expressed in the thymus of a number of vertebrates from mice to cartilaginous fish [102,103]. In sharks, the variable regions of TCRα, γ, δ chains are modified by hypermutation, contributing to the diversity of TCR recognition [102].

Intracellular levels of AID also are highly regulated. AID is primarily localized to the cytosol, particularly in association with the heat-shock protein Hsp90, eEF1A, and Hsp40-DnaJa1 [104,105,106]. After B-cell activation in germinal centers, AID is translocated to the nucleus, where SHM of immunoglobulin heavy and light chains occurs [107]. Other cellular genes are edited at a much lower frequency, including c-*Myc* and *Bcl6* [107]. AID degradation occurs rapidly in the nucleus via a CUL7 E3 ubiquitin ligase in complex with FBXW11 [108], or AID is shuttled back to the cytoplasm using CRM1 [109] to control its mutational activity. Editing by AID is associated with DNA demethylation, the increased expression of repetitive elements, such as long and short interspersed nuclear elements (LINEs and SINEs, respectively), as well as lymphoma induction [110]. Therefore, AID exemplifies the innate immune role of the APOBEC family to control the spread of viruses and transposons, but it also affects developmental processes that lead to adaptive immunity and control of autoimmune activity.

## 2. Viral Restriction by APOBEC Enzymes

The ability of APOBECs to inhibit viral replication was first noted during studies of HIV-1 that involved a lack of expression of the viral infectivity factor (Vif) encoded by a gene originally known as *sor* [111]. Vif-minus HIV-1 replicates well in certain T-cell lines that are “permissive”, whereas other “non-permissive” lines as well as primary T cells restricted Vif-mutant HIV-1 [112]. Using a complementary DNA subtraction screen of related CEM permissive and non-permissive T-cell clonal lines, Sheehy et al. isolated a cellular gene called CEM15, which could restrict Vif-minus HIV-1 in permissive cells [113]. CEM15 was later found to be a member of the APOBEC family, A3G [114]. Many subsequent studies have confirmed the ability of APOBEC proteins to restrict both RNA- and DNA-containing viruses. In the following sections, we will discuss APOBEC inhibition of virus replication through deaminase-dependent and deaminase-independent mechanisms [1,36,53].

### 2.1. Deaminase-Dependent Inhibition of Viruses by APOBECs

#### 2.1.1. HIV-1 Hypermutation by A3 Enzymes in Cell Culture

Early experiments indicated that similar quantities of HIV-1 virions were produced by non-permissive cells in the presence and absence of Vif. However, the infectivity of Vif-minus virions was greatly decreased as a result of A3G packaging [115,116]. Infection of cells with A3G-containing HIV-1 resulted in C-to-U deamination of ssDNA minus strands during reverse transcription of the viral RNA genome. Completion of proviral DNA synthesis resulted in G-to-A mutations of plus strands [1,53], and ≥10% of G residues were mutated [117]. Among the A3 family, A3G is expressed at higher levels than other A3 family members in T cells, resulting in the majority of HIV-1 mutagenic activity [118]. However, A3F and A3H contribute lower amounts of viral DNA deamination, with even lower levels by A3D [118]. A3B and A3C are weakly or moderately expressed in T cells, but are more likely to contribute to simian immunodeficiency virus (SIV) mutagenesis [119]. A3B and A3C, together with A3G and A3F, provide a high barrier for SIV infection of humans [119]. Thus, the level of A3 restriction is proportional to enzyme expression and packaging into virions. This conclusion is supported by the observation that infected patients with higher A3G expression have lower levels of circulating HIV-1 RNA and disease progression [120,121,122,123].

The prevailing mechanism for reduced HIV-1 infectivity mediated by A3 enzymes has been shown in cell culture studies [1,124,125] as well as experiments in HIV-infected humanized mice [126,127] (Figure 2). In the absence of Vif, cytosolic A3 proteins bind HIV-1 genomic RNA, cellular RNA, and HIV-1 nucleocapsid (NC) protein for assembly into virions [128,129,130,131,132,133]. NC binding is an essential step for A3 virion packaging. For efficient packaging, the interaction between NC and APOBEC is highly dependent on APOBEC association with diverse RNAs, including viral RNA and cellular RNA [130,131,132,133]. Multiple studies have demonstrated that 7SL1 non-coding RNA (ncRNA) may play a more significant role than the HIV-1 genomic RNA in A3 assembly into virions [36,48,49,50,134]. A unique feature of A3G is its binding to the Alu region of 7SL1 RNA for virion packaging [51,133,135,136]. A comparative study using cross-linking immunoprecipitation coupled to next-generation sequencing has shown that A3 proteins bind HIV-1 genomic RNA preferentially over cellular RNA in infected cells [51]. In immature and mature virions, A3 proteins bind G- and A-rich RNA sequences, mimicking properties of the NC domain of the HIV-1 Gag protein, which is a major constituent of immature particles [51,137]. These data suggest a model in which A3 incorporation into HIV-1 virions is facilitated by preferential binding to G- and A-rich RNA [51,137]. Each virion contains approximately 7 (±4) A3G proteins that are associated with HIV genomic RNA in the virion core [138,139]. However, 1–2 A3G per virion may be sufficient for HIV-1 hypermutation and restriction [140].

Reverse transcription occurs after virion entry and uncoating. During reverse transcription, RNase H-dependent degradation of the HIV-1 RNA template exposes newly synthesized ssDNA for A3G binding. A3G must disengage from HIV-1 genomic RNA to bind nascent minus strand reverse transcripts for deoxycytidine deamination preferentially in the 5′ CC 3′ context [38,40,141]. Moreover, the frequency of deamination increases 5′ to 3′ across the HIV-1 proviral genome, perhaps due to the duration of the ssDNA state during reverse transcription [112,142]. A3G oligomerization is necessary to support deamination activity [40,141]. RNase H digestion is also required for A3H-mediated deamination, but a small RNA duplex remains bound to purified A3H after RNase treatment. This RNase-insensitive RNA duplex supports A3H dimerization and deamination on DNA [44,45,46]. RNA-mediated A3H dimers from pig-tailed macaques [47] and chimpanzees [143] stabilized enzyme activity and prevented proteasomal degradation. As noted previously, multimerization and RNA binding are common features of APOBEC activity.

Deamination-induced uridines in retroviral complementary DNA (cDNA) can be subsequently recognized by the uracil-DNA glycosylases (UDGs) in the uracil base excision repair (UBER) pathway [144], potentially leading to base removal. Recognition by apurinic/apyrimidinic endonucleases (APE1 or APE2) at abasic sites leads to DNA cleavage and even degradation [144]. Decreased levels of the UDG and uracil N-glycosylase 2 (UNG2) enzymes have been shown to reduce A3G antiviral activity [144,145]. Mutation gradients are observed 5′ of the central and 3′ polypurine tracts [146,147]. More recently, deep sequencing experiments using Vif-deficient HIV-1-infected T cells examined mutations within minus-strand strong-stop DNA in the presence of A3G [148]. Five hotspots corresponding to consensus A3G deamination sites were bound by UNG2 to create abasic sites and subsequent cDNA cleavage by cellular endonucleases. Endonucleolytic cleavage was incomplete since substantial proportions of cytidine-to-uridine edited cDNAs were observed and remained intact. These studies also suggested that UBER processing of edited HIV-1 DNA acts as a pathogen-associated molecular pattern (PAMP) that would elicit an innate immune response [148].

Different A3 (A3G, D, F, H) enzymes are all capable of inducing C-to-U deamination in HIV-1 ssDNA. A3G has a major role in hypermutation in T cells and macrophages [118,149]. Although different A3 enzymes can be co-encapsidated into the same virion [150], endpoint proviral DNA analysis from patients showed that these deaminases rarely functioned on the same genome [151]. Several studies showed a synergistic effect of multiple A3 enzymes, in which additional mutations or restriction provided more than an additive effect [150,152,153]. Recent experiments [154] demonstrated that, in a single cycle infection, A3F and A3G could induce a combined increase in HIV-1 DNA mutations and synergistically reduced HIV infectivity. A3F and A3G could form hetero-oligomers in the absence of RNA, and this mixed oligomer promoted A3G processivity and increased deamination. The hetero-oligomer also decreased the efficiency of HIV-1 reverse transcriptase, providing additional time for A3-catalyzed deamination [154]. A subsequent study from the same group demonstrated that A3F haplotypes with different polymorphisms could all form hetero-oligomers with A3G [155]. The A3F 231V variant formed a more stable hetero-oligomer with A3G, leading to increased resistance to Vif-mediated A3 degradation and provided increased HIV-1 restriction [155]. These studies indicate that different A3s cooperate to reduce HIV replication but must overcome Vif-mediated antagonism.

In addition to the G-to-A hypermutation induced during reverse transcription, the deamination activity of A3G has been shown to interfere with early HIV-1 transcriptional activity (Figure 2) [156]. During reverse transcription, the 5′ end of viral RNA is degraded by RNase H to yield minus-strand strong-stop DNA prior to the first strand transfer. This strong-stop DNA is a ssDNA target for A3G. After completion of reverse transcription, transcription by RNA polymerase II yields a transactivation response (TAR) element consisting of a short stem–loop RNA structure essential for viral RNA elongation, in which two different cytidine residues are the substrate for A3G [34,157]. One A3G deamination converts the G in 5′-CTGGGA-3′ to A on the proviral plus strand to disrupt the TAR loop and binding of cellular factors, such as pTEFb. Viral transcription elongation is inhibited, resulting in the accumulation of short viral transcripts that reduce HIV-1 replication [156]. The combination of proviral hypermutations by diverse A3 multimers to generate defective proviruses with decreased full-length viral transcription ultimately provides different deaminase-dependent mechanisms to inhibit virus production.

#### 2.1.2. A3-Mediated Hypermutation in HIV-1-Infected Patients

Although hypermutation of HIV-1 proviruses by A3 deaminases has been widely studied in cell lines, there is ample evidence of A3 activity in infected patients. Patient samples showed G-to-A hypermutation of proviruses recovered from HIV-1-infected peripheral blood mononuclear cells (PBMCs). Although many normal proviruses were recovered, 43% of the mutant proviruses had in-frame stop codons and non-synonymous nucleotide changes [158]. High A3G activity has been correlated with slower disease progression, whereas lower A3G activity resulting in sub-lethal mutants likely enables HIV diversity and rapid disease progression [159]. Most recent studies suggest that A3 expression has both a beneficial and a harmful influence on HIV-1 infection in vivo [159,160,161,162,163].

The beneficial role of A3-mediated mutations for in vivo HIV-1 infection has been documented in multiple studies. Results showed that deaminase-generated variability of HLA-recognized viral epitopes allowed escape of virally infected cells from cytotoxic CD8+ T cells [160,164,165,166]. The most frequently A3G-targeted motifs had higher numbers of non-synonymous mutations within *gag*, *pol*, and *nef* sequences encoding immunodominant CD8+ T cell epitopes [166]. CD8+ T cells from infected individuals showed diminished responses against these mutant viruses [165]. Deep sequencing of viral DNA from HIV-1-infected patients also revealed G-to-A mutations in these preferred known/predicted epitopes [160]. However, experiments also showed that this immune escape mechanism depended on the HLA-class-I allele expressed in patient CD8+ T cells [164]. For HLA-B57 and A2-restricted epitopes, the APOBEC-induced mutations diminished the recognition of these epitopes by CD8+ T cells, but for HLA-B35-restricted epitopes, the APOBEC mutation increased CD8+ T-cell recognition. The HLA-B35-restricted epitopes elicited the lowest level of CD8+ T-cell engagement, and significantly fewer APOBEC-mutation hotspots were detected in viral genomic sequences encoding HLA-B35-restricted epitopes than other HIV-1-encoded epitopes. Thus, the immune recognition generated by this epitope likely allowed virus survival [164]. A recent bioinformatics analysis also revealed that A3 mutational hotspots in the HIV-1 genome encoded epitopes that most often resulted in diminished HLA binding, suggesting a co-evolution of the viral genome with deaminase activity [166].

The lethal effect of A3G-induced HIV-1 hypermutations generated in vivo was shown by analyzing viral polymerase sequences from 3000 chronically infected patients [167]. In this study, G-to-A mutations occurred more often in A3G-disfavored compared to A3G-favored motifs, whereas mutations in A3F-disfavored contexts were infrequent. Nucleotide changes frequently occurred in A3F or other A3-favored contexts. Analysis of HIV-1 envelope sequences from patients in acute or early infection stages also showed this preference [167]. These results suggest that A3G-induced mutagenesis is lethal to HIV-1, but mutagenesis caused by A3F and/or other deaminases may result in non-synonymous mutations that promote viral diversification [167]. Viral sequences were isolated from elite controllers who restrict HIV-1 viremia without antiviral treatments. These sequences revealed attenuated Vif activity against A3G in cell culture-based assays [123]. Furthermore, variant A3G proteins expressed in HIV-1-infected individuals, e.g., A3G-H186R, had lower antiviral activity [168], and patients homozygous for this allele had higher viral loads and rapid disease progression [168,169,170,171,172]. These studies support the idea that A3G leads to lethal hypermutation in vivo, whereas other A3 proteins may provide mutations that allow HIV-1 selection and survival.

Chronically infected HIV-1 patients are known to harbor numerous defective proviruses and a reservoir of virus-positive cells [173]. Open questions include whether APOBECs play an important role in generating this viral reservoir and the timing of its establishment. Using deep sequencing technology to examine HIV-1 proviruses from four patients at regular intervals, a recent study found that most HIV-1 DNA genomes remained intact early in the infection process. Viral genomes hypermutated by A3G or A3F were not observed during the earliest period of infection, yet increased over time. Single-base substitution mutations in proviral genomes during the first year after infection and the lack of truncated proviruses revealed errors consistent with those generated during reverse transcription. Despite the limited sample size, these experiments suggest that A3-mediated changes accumulate during the chronic phase of HIV-1 infection [174].

#### 2.1.3. Vif Antagonism of A3 Proteins

HIV-1 encodes the Vif protein to inhibit the function of A3s to prevent hypermutation and allow efficient virion replication [111,175,176,177,178,179,180]. The well-accepted inhibitory mechanism is that Vif promotes the formation of an E3 ubiquitin ligase complex to target A3s for proteasomal degradation (see Figure 2). Through heterodimerization with the transcription co-factor, core-binding factor beta (CBFβ), Vif recruits the cullin–RING E3 ligase 5 (CRL5) complex, which includes cullin5 (CUL5), Elongin B (ELOB), Elongin C (ELOC), and RING-box protein 2 (RBX2), to polyubiquitinate E3 proteins prior to degradation by the 26S proteasome [114,142,181,182,183,184,185]. CUL5 provides a scaffold for complex assembly, whereas ELOB/ELOC and RBX2 serve as adapters to Vif and an E2, respectively [183,186]. Interestingly, Vif sequestration of CBFβ provides a “dual hijacking mechanism” since CBFβ is limiting for interaction with RUNX1, which reduces its activity as a transcription factor [187]. Ubiquitin-like modifier NEDD8 conjugation to cullins (neddylation) is required for a conformational change and assembly of functional cullin E3 ligases that are needed for substrate ubiquitylation [188]. Proteasomal degradation of polyubiquitinated A3 prevents their packaging into virions, which is necessary for inhibition of virus replication (Table 1).

Recent studies have provided additional details of Vif antagonism of E3 enzymes. A RING-finger E3 ubiquitin ligase (MDM2) downregulates Vif through proteasomal degradation [189,190]. Co-immunoprecipitation experiments showed that CBFβ interaction with Vif prevents MDM2 binding, leading to Vif stabilization. Mutation of the MDM2-binding residue of Vif (R93E) prevented MDM2-mediated degradation, and HIV-1 containing this mutation was more resistant to A3G [191]. Additional experiments have shown that Vif interaction with a RING-between-RING (RBR) E3 ubiquitin ligase (ARIH2) is essential for CRL5-dependent HIV infectivity in primary CD4+ T cells [187]. In a “tag-team” mechanism, Vif-CBFβ together with the CRL5 ubiquitin ligase complex recruits ARIH2 to directly transfer the first ubiquitin from an E2-conjugating enzyme to A3G. This initial ubiquitylation then accelerates polyubiquitylation by UBE2R1, but is not essential for A3G ubiquitylation [187]. Ubiquitylation occurred on lysine residues throughout A3G N- and C-terminal regions [192,193]. Although both N- and C-terminal regions of A3G can be polyubiquitylated, the N-terminal domain contributes less to Vif-mediated proteasomal degradation [193]. Consequently, substitutions of lysine to arginine in the A3G C-terminal domain increased A3G resistance to Vif and blocked HIV-1 replication [193]. Thus, HIV-1-Vif degradation of A3 proteins is regulated in multiple ways by interactions with the ubiquitylation machinery.

A second mechanism for Vif-mediated antagonism occurs through restricted translation of A3G mRNA. Vif inhibits the translation of A3G mRNA to further decrease its intracellular level in HIV-1-infected cells. Subsequently it prevent A3G packaging into virions [194,195,196,197]. The 5′ untranslated region (UTR) of A3G mRNA has two stem–loop structures that are required for Vif to inhibit A3G translation. Vif may block ribosomal scanning at the 5′ UTR of A3G mRNA, although the binding site for Vif in this RNA region has not been identified [195,197]. A degradation-deficient Vif mutant (K26R) retained the ability to inhibit A3G translation [197]. In contrast, the H42/43N Vif mutant had decreased association with A3G and no effect on A3G translation. Translational inhibition by Vif partially restored viral infectivity in A3G-expressing cells in the absence of proteasomal degradation, consistent with a degradation-independent mechanism [197]. These results suggest that HIV-1 has evolved several mechanisms specifically to counteract A3G antiviral activity (Table 1).

Although Vif decreases the level of A3G by several means to prevent its antiviral activities, cells have developed several ways to counteract Vif and stabilize A3G. First, cellular proteins may directly interrupt A3G degradation by Vif [198,199]. A host deubiquitinating enzyme (DUB), USP49, was shown to directly bind A3G and remove the K48-linked ubiquitin on the deaminase to inhibit Vif-mediated A3G degradation. In primary CD4+ T cells from HIV-1-infected individuals, a strong correlation was observed between USP49 and A3G expression levels. A negative correlation between plasma HIV-1 RNA levels and USP49 expression was also noted [198]. Another study showed that a MAP3 kinase, the apoptosis signal-regulating kinase 1 (ASK1), binds Vif to disrupt the assembly of Vif-ubiquitin E3 ligase complexes, which stabilizes A3G to promote virion incorporation and infectivity. This Vif counteraction mechanism was further enhanced by treating infected cells with the antiretroviral drug AZT to induce ASK1 expression [199]. The mechanism for AZT induction of ASK1 does not involve ASK1 kinase activity. Another HIV-1 accessory protein, Nef, interacts with ASK1 to thwart apoptosis of infected cells [200]. However, whether there is competition between Nef and Vif for ASK1 during HIV-1 infection remains unclear.

A3G levels also are maintained through recruitment of cellular proteins to reduce Vif levels [201,202]. Cyclin F protein directly binds Vif through an RKL motif (amino acids 167–169). This interaction then makes Vif a substrate for the SCF^cyclin-F^ E3 ligase complex, leading to proteasomal degradation. Consequently, cyclin F stabilizes A3G levels, which increases virion incorporation and decreases infectivity. However, in primary HIV-1-infected T cells, cyclin F levels are downregulated, although the mechanism is unknown [202]. The cellular histone deacetylase 6 (HDAC6) also directly interacts with A3G to compete for Vif-A3G interactions, neutralizing Vif-mediated A3G ubiquitylation. HDAC6 interacts with Vif to promote Vif autophagic degradation, which is dependent on the bound-to-ubiquitin-zinc finger (BUZ) domain of HDAC6 and its deacetylase activity [201].

Together, these studies suggest that HIV-1 maintains a fine balance between Vif-mediated destruction of A3 enzymes and the ability of the host to resist viral infection. A3G-induced mutations allow rapid virus selection, particularly in the face of high interferon levels or antiviral drugs. Although reverse transcription allows basal levels of mutation, mutagenesis by A3 enzymes provides viruses with the ability to respond immediately to changes in the host environment.

Other lentiviruses counteract APOBECs by different mechanisms. The Vif protein from the poorly pathogenic feline immunodeficiency virus (FIV) subtype B only weakly antagonizes feline A3 proteins. Instead, the viral protease cleaves feline A3 proteins in released virions [203] (Table 1). This study provides the first evidence that lentiviruses encode two anti-A3 factors [203]. In addition, passage of HIV-1 isolates with irreparable Vif deletions developed substitutions of amino acid residues in the Env proteins of A3G-resistant viruses. These Env adaptations decreased virus fusogenicity, allowing higher levels of Gag-Pol packaging into the virions. Higher Pol levels yielded faster reverse transcription to protect the virus genome from A3G-mediated hypermutation [204]. Another HIV-1 accessory protein, viral protein R (Vpr) binds to A3G and downregulates deaminase activity through Vpr-binding protein (VprBP)-mediated proteasomal degradation. A3G encapsidation in progeny virions subsequently is reduced [205]. Therefore, these studies suggest that different viral proteins may cooperate to counteract the antiviral functions of APOBECs.

#### 2.1.4. Deamination-Dependent Hypermutation of HTLV-1

Human T-Cell Leukemia Virus type 1 (HTLV-1) is a human deltaretrovirus that causes both Adult T-Cell Leukemia/Lymphoma (ATL) and HTLV-1-Associated Myelopathy/Tropical Spastic Paraparesis (HAM/TSP) [206]. Like HIV-1, HTLV-1 targets T lymphocytes, which express A3 enzymes [121]. Nevertheless, HTLV-1 is more resistant to hypermutation relative to HIV-1 [62,207,208,209]. Hypermutated HTLV-1 genomes were not identified in peripheral blood mononuclear cell DNA from ten HTLV-1 carriers [208]. Genomic DNA from carriers revealed G-to-A nonsense mutations at low frequency (0.21% and 0.11% for the *pol* and *tax* genes, respectively) typical of A3G preferred motifs, although other deaminases, such as AID, were implicated. No significant correlation was observed between A3G expression levels and the number of G-to-A mutations in proviral genomes [207]. However, HTLV-1 has an accessory protein, the HTLV-1 basic leucine zipper factor (HBZ), which is encoded on the proviral minus strand and is required for the maintenance of HTLV-1-transformed cells. Interestingly, the *HBZ* gene lacks A3G-targeted sequence motifs or evidence of A3 mutation, consistent with selection for its function [207].

Although HTLV-1 hypermutation is rare in vivo, a small percentage of genomes (0.1–0.5%) are edited extensively, and up to 97% of cytidines in the edited genomes are deaminated [208]. Cell culture experiments showed that both overexpressed and endogenous A3G proteins were packaged into HTLV-1 virions and inhibited HTLV-1 infection without overt cytidine deamination activity [210]. Additional cell culture studies showed that A3A, A3B, and A3H hapII were packaged into HTLV-1 virions, and A3A and A3B restricted HTLV-1 in a deamination-dependent manner. G-to-A mutations on the viral plus strand induced by A3A and A3B occurred in a GA to AA dinucleotide context, whereas A3H hapII acted independently of deamination activity. Proviral DNA isolated from different T-cell lines generated from HTLV-1-infected patients showed G-to-A mutations from A3A and A3B as well as A3G [211]. A recent study using humanized mice revealed that A3B expression increased following HTLV-1 infection, but the mutation profile was not investigated in this study [212]. These studies agree that APOBEC enzymes, primarily A3G, lead to lower levels of HTLV-1 mutation relative to those observed for HIV-1.

Two reasons have been proposed to explain the relative resistance of HTLV-1 to APOBEC-mediated hypermutation [213]. First, although HTLV-1 does not encode a Vif-like homologue to degrade APOBECs, a peptide motif in the C-terminal domain of the HTLV-1 nucleocapsid (NC) protein acts in *cis* to inhibit A3G packaging. Thus, A3G packaging efficiency into nascent virions is reduced compared to HIV-1 particles made in the absence of Vif [209]. Second, HTLV-1 and HIV-1 employ different replication strategies. After primary infection, HTLV-1 maintains a low level of productive replication that is amplified mainly through oligoclonal expansion of infected cells. In contrast, active HIV-1 replication involves a high rate of de novo infection and cell death [214,215,216,217]. Therefore, infrequent replication through reverse transcription leads to relatively few opportunities for APOBECs to cause mutations within the HTLV-1 genome [213]. A3 mutagenesis of HTLV-1 genes encoding immunogenic proteins needed for viral replication, such as Tax, provides survival value against elimination by cytotoxic T cells, while avoiding inactivation of the *HBZ* gene needed for proliferation of HTLV-1-infected cells [207] (Table 1).

#### 2.1.5. Deamination-Dependent A3 Activity and Murine Retroviruses

The first in vivo evidence indicating that A3 proteins protect their hosts from retrovirus infection was obtained using the betaretrovirus Mouse Mammary Tumor Virus (MMTV) [218]. Unlike primates, mice have a single A3 protein (mA3). Infection of mA3-knockout mice on the B6 background with the RIII MMTV strain showed increased virus production and spread relative to infections of wild-type mice. This study also showed that mA3 was packaged into virions. Deaminase packaging was dependent on the MMTV nucleocapsid protein and viral RNA. However, unlike HIV-1, cytidine deamination of the MMTV genome was not observed, suggesting that mA3 has a deaminase-independent mechanism for viral inhibition (see Section 2.2).

In contrast, our experiments with two different MMTV strains that lack expression of the Rem protein showed APOBEC-mediated deamination in vivo [219]. Rem is a doubly spliced version of the envelope mRNA and is expressed as a truncated in-frame version of MMTV Env. Elimination of a splice donor (SD) site enabled construction of a mutant MMTV proviral clone that prevented Rem production (MMTV-SD), yet did not affect viral replication in cell culture [220]. Injection of MMTV-SD and wild-type MMTV (MMTV-WT) independently into BALB/c mice showed that loss of Rem expression reduced proviral loads and mammary tumor incidence. Analysis of proviruses from MMTV-SD-induced mammary tumors revealed frequent reversion of the SD mutation by recombination with endogenous *Mtv* sequences [220], indicating selection in vivo.

To determine whether selection of proviruses occurs due to mutation by APOBEC deaminases, we performed sequence analysis of the *env* gene from proviral DNA isolated from MMTV-SD and MMTV-WT-induced tumors. Both transition and transversion mutations were increased in MMTV-SD proviruses relative to MMTV-WT proviruses. We observed that G-to-A mutations on the plus strand were elevated in the absence of Rem expression, yet C-to-T mutations showed the largest increase on both proviral strands [219]. Human A3 and mA3 preferentially induce G-to-A hypermutations on HIV-1 plus strands after C-to-U deamination of minus strands during reverse transcription [221,222]. However, the APOBEC family member AID acts on both DNA strands, leading to G-to-A and C-to-T mutations on the coding strand of the variable regions of immunoglobulin genes [107]. AID is expressed primarily in B cells, whereas mA3 is expressed in multiple tissues, including T cells. MMTV is known to require replication in B and T cells as well as dendritic cells prior to mammary gland transmission, exposing the virus to inhibition by APOBECs [218]. Therefore, APOBEC enzyme activities in multiple cell types potentially restricts MMTV spread and viral loads in the absence of Rem.

Sequence motif analysis was performed to determine the identity of APOBEC enzymes targeting MMTV proviruses lacking Rem expression. Significant mutations of the 5′-WRC-3′ motif, a known “hotspot” for AID hypermutation [223], were observed in MMTV-SD proviruses compared to MMTV-WT proviruses [219]. However, we also detected increased mutations in 5′-TYC-3′ and 5′-ATC-3′ motifs, which are associated with mA3 activity, in MMTV-SD proviruses [221,224,225], as well as less frequent 5′-SYC-3′ mutations known as AID “coldspots” [226,227]. Furthermore, proviral C-to-T mutations were significantly reduced in MMTV-SD-induced tumors from *Aicda*^−/−^ (AID-knockout) mice compared to wild-type BALB/c animals. Low-level proviral C-to-T mutations were not significantly different in tumors from AID-knockout and wild-type mice when the inoculated virus expressed Rem [219]. Surprisingly, mutations within proviral 5′-WRC’3′ and 5′-TYC-3′ motifs (typical of AID and mA3, respectively) declined in *Aicda*^−/−^ mice. These data suggest that MMTV-encoded Rem is a Vif-like antagonist of mutations induced by AID, and perhaps other APOBECs, during viral replication in vivo [219] (Table 1).

One potential problem with this interpretation is that the splice donor mutation within MMTV-SD also eliminates superantigen (Sag) expression from an internal viral promoter used in lymphocytes [220]. MMTV Sag is an accessory protein required for viral transmission in vivo. To confirm the involvement of Rem, not Sag, in APOBEC-mediated hypermutation, we used a Sag-independent MMTV strain, type B leukemogenic virus (TBLV), to induce T-cell lymphomas in BALB/c mice [228]. Like tumors induced by MMTV-SD lacking Rem expression, reduced proviral DNA loads were observed in tumors induced by inoculation of a Rem-minus virus (TBLV-SD). High throughput Illumina sequencing revealed significantly elevated G-to-A changes on the plus strand of TBLV-SD proviruses compared with TBLV-WT proviruses. The WRC, SYC, and TYC-motif mutations were significantly increased in the TBLV-SD proviruses that corrected the splice donor mutation by recombination with endogenous *Mtvs*. Since TBLV is not transmitted to the mammary gland, selection for Rem expression likely occurs during viral replication in lymphoid cells. These results are consistent with the role of Rem as an APOBEC antagonist [219].

To determine whether Rem functions similar to HIV-1 Vif by targeting A3 enzymes for degradation, we co-expressed either AID or mA3 in the presence or absence of Rem. Rem expression alone or in the context of an intact provirus led to proteasome-dependent degradation of AID, but not mA3. In addition, as previously shown, mA3 was packaged into virions [218]. Our results revealed that AID is not packaged, and Rem does not affect virion incorporation of either mA3 or AID [219]. The mechanism of Rem antagonism remains under investigation.

Like the betaretrovirus MMTV, gammaretroviruses are restricted by APOBEC proteins. AKV, an endogenous, ecotropic murine leukemia virus (MLV) strain, which also replicates as an exogenous virus, has been shown to be hypermutated by mA3 [221,229,230]. Overexpression of mA3 in cell culture experiments indicated that the AKV genome was mutated more significantly than Moloney MLV (M-MLV). Splenocytes isolated from *mA3*-knockout mice and infected with AKV also showed higher infectivity than virus-infected splenocytes from *mA3*^+/+^ or heterozygous *mA3*^+/−^ mice [221]. More recent studies demonstrated that endogenous mA3 levels in NIH3T3 cells induce G-to-A hypermutations in AKV transcripts and restrict viral replication [230]. Analysis of over 1,000 singly infected NIH3T3 cells by Western blotting revealed that only 20% of the cells expressed detectable mA3 levels. These results correlated with the percentage of hypermutated viral transcripts. Analysis of the sequence context of hypermutations indicated that deamination occurred on the proviral minus strands in the conserved mA3 motif 5′-TTCAA-3′ [230]. Proviral DNA analysis from B6 mice also showed G-to-A or C-to-T hypermutations in an mA3-deamination context (5′-CC-3′ or 5′-TC-3′) [229].

Several studies have shown that gammaretroviruses make a glycosylated translation product of the *gag* gene. This product, glyco-Gag (gGag), uses a CUG initiation codon upstream and in the same frame as the Gag polyprotein [231,232,233] for antagonism of mA3 restriction activity [229,234,235,236] (Table 1). Alternative translation of the *gag* reading frame results in 88 additional amino acids at the Gag N-terminus to give a signal peptide that allows ER insertion and gGag Pr85 production [237]. Pr85 gGag is a type I transmembrane precursor that is further processed by proteases to generate a 55-kDa N-terminal fragment containing a leader peptide, matrix (MA), and the p12 protein. A C-terminal fragment containing the viral capsid (CA) and nucleocapsid (NC) proteins is also generated and secreted [238,239]. The N-terminal product is a type II integral membrane protein at the cell surface and is incorporated into virions. Incorporation of the 55 kDa protein is required for efficient virion budding and release [238,239,240,241]. The AKV gGag has been shown to counteract G-to-A mutations induced by mA3, but has only two N-glycosylated residues compared to three sites found in M-MLV gGag [229]. The number of gGag glycosylation sites correlated with the level of viral resistance to deamination, suggesting the importance of post-translational modifications in restricting A3 activities [229]. However, mechanistic details of gGag glycosylation and regulation of AKV mutagenesis by mA3 remain unknown.

Although the role of mA3 has been the primary focus of research on murine retroviruses, other APOBEC proteins also have been studied. Mouse APOBEC1 (mA1) was first shown to restrict MLV, but a second study contradicted these results [242,243]. Compared to human A1, which exhibits restricted expression in the small intestine, mA1 also is highly expressed in the liver, spleen, bone marrow, and lymph nodes [244,245,246]. Since in vitro experiments showed that human A1 deaminates DNA substrates [21,22,23], it is possible that mA1 functions as a restriction factor for MLV. The first study from Petit et al. [242] suggested that mA1 induced C-to-T and G-to-A mutations in Friend MLV (F-MLV) transcripts, and mA1 preferentially mutated the TC context, whereas mA3 preferentially mutated in the CC dinucleotide context in vitro. The investigators also obtained F-MLV sequences from spleens of infected B6 wild-type mice. TC motif mutations in these transcripts were interpreted as mA1 mutagenic activity that restricts F-MLV in vivo [242].

Subsequently, several groups demonstrated that mA3 also edits cytidines in the TC motif [221,222,224,225]. One study [243] addressed whether mA1 restricts F-MLV by infection of mA1-knockout mice. Inoculation of F-MLV into wild-type or mA1-knockout mice showed no differences in infection or plasma viremia levels. Further, proviral DNA sequence analyses failed to show significant differences in G-to-A mutations between wild-type and mA1-knockout mice. However, mice lacking a functional *mA3* gene demonstrated lower numbers of GA to AA mutations compared to wild-type or mA1-knockout mice, suggesting that mA3 targeted the F-MLV reverse transcripts in the TC, but not the CC context. Since mA1 could potentially edit retroviral RNA, this study also analyzed the percentage of F-MLV TC-TT and CC-CT mutations within viral transcripts. No differences in dinucleotide mutation frequencies were observed between viral RNA obtained from mice with or without a functional *mA1* gene. Therefore, this study indicated that mA1 does not restrict F-MLV replication in vivo.

#### 2.1.6. Deamination of Other RNA-Containing Viruses by APOBECs

The retroviral subfamily *Spumavirinae* also are restricted by APOBECs. The foamy viruses use Bet protein to sequester A3 proteins away from the virus assembly site and prevent their incorporation into virions [247,248,249] (Figure 2). A3 editing of Bet-deficient feline foamy virus (FFV) resulted in greatly decreased FFV titers [247]. In contrast to lentiviruses, cytidine deamination occurred in A3-positive FFV-producing cells [247]. Unlike Vif activity on several A3 proteins, Bet from multiple foamy viruses (e.g., FFV and Prototype Foamy Virus/PFV) does not induce A3 degradation. Instead, Bet binds A3G to block dimerization [247,248,249,250,251] and sequester deaminases in insoluble complexes that are unavailable for virion packaging (Table 1). Alternatively, Bet alters A3C subcellular localization to be exclusively cytosolic [250,251]. Simian Foamy Virus (SFV) is transmitted from non-human primates to humans, but not between humans [252,253,254,255]. Bioinformatic analysis showed that human A3G caused distinct hypermutations within specific dinucleotide contexts in the *gag* gene of the SFV genome [256]. In human hosts, hypermutation occurred more frequently in GG and GR (i.e., GG or GA) motifs, whereas macaques had a relative increase in GA and GM (i.e., GA or GC) motif mutations. More stop codons were also identified in SFV sequences from humans relative to monkeys. These data suggest that hA3G induces mutations to protect against SFV replication and to prevent virus transmission between humans [256].

APOBECs also restrict other RNA-containing viruses such as coronaviruses, measles viruses, and enteroviruses 71 (EV71) [257,258,259,260,261]. Although editing of HIV-1 by APOBECs was identified in 2004 [262], this restriction mechanism is not common among other RNA-containing viruses. However, a notable characteristic of coronaviruses is their highly U/A-rich and C/G-poor genome. For example, the human coronavirus strain HCoV-NL63, which causes the common cold in healthy adults, but severe symptoms in young, elderly, and immunocompromised individuals [263,264,265], has 39% U and 27% A content, with only 14% C and 20% G nucleotides [263]. This bias suggests that the virus has been selected by cytidine deamination during evolution.

To test this idea, Milewask et al. [259] infected human airway epithelial cells with HCoV-NL63. Upregulation of A3A, A3C, A3D, A3G, and A3F transcripts was observed, with an almost 100-fold increase in A3A expression. Replication of HCoV-NL63 was also reduced by 1.5 to 2 logs in other permissive cells (LLC-Mk2) overexpressing A3C, A3F, and A3H. However, A3A, A3D, or A3G overexpression did not inhibit virus replication. Transfection of the catalytic mutants of A3C, A3F, or A3H produced lower inhibition than the wild-type A3s, suggesting that A3 antagonism of the NL63 coronavirus was deamination-dependent (Table 1).

To determine whether A3 proteins cause hypermutation of the viral genome, recombinant GC-rich HCoV virus strains were modified with GC-rich potential “hot-spots” for cytidine deamination. No enrichment of G-to-A or C-to-T mutations was observed in coronaviruses containing these GC-rich regions after infection of cells expressing different A3s. Sequence heterogeneity of coronaviruses was detected in infected cells expressing A3F. Both wild-type and catalytic inactive mutants of A3C, A3F, and A3H interacted with the HCoV-NL63 nucleocapsid (N) protein, but not other A3 proteins (A3A, A3D, and A3G). Colocalization also was observed between A3C and NL63 N proteins, suggesting that APOBEC proteins restrict coronavirus infections [259]. Further, SARS CoV-1 nucleocapsid (N) protein bound to human A3G in the presence of RNA and viral M protein to allow A3G packaging into viral-like particles [266]. More recent studies have shown mutational signatures of APOBECs or Adenosine Deaminase Acting on RNA (ADAR) on SARS-CoV-2 genomes obtained from archived sequences of viral isolates or patient samples [267,268]. The ADAR enzyme has been shown to act on double-stranded RNA, which is encountered during replication of RNA-containing viruses [269]. Cytosine-to-uracil changes typical of APOBEC deaminases were observed with a lower frequency than the adenosine-to-inosine changes typical of ADARs. The C-to-U changes were biased toward the viral plus strands and primarily occurred downstream from uridines and adenosines [268], a sequence context documented for A1-mediated deamination [13,15]. Such results suggested that APOBECs and the ADARs have driven the evolution of coronaviruses since 65% of all coronavirus mutations could be attributed to these enzymes [267]. Overall, additional studies are needed to clarify APOBEC editing of SARS coronaviruses as well as how CoVs may antagonize these deaminases.

#### 2.1.7. Deamination-Dependent Antagonism of DNA-Containing Viruses

Epstein-Barr Virus (EBV) belongs to the gammaherpesvirus subfamily and infects B cells to induce infectious mononucleosis and multiple tumors, such as Burkitt lymphoma [270]. In EBV-infected B cells, AID is activated and induces cytosine deamination and somatic hypermutation [271]. Like EBV, another gammaherpesvirus, Kaposi’s sarcoma herpesvirus (KSHV) or HHV-8, replicates in B cells, which is the primary site of AID expression. Viral infection of B cells activates AID as well as surface expression of NKG2D ligands through DNA damage detection, targeting them for elimination by natural killer (NK) cells. To avert this effect, KSHV encodes at least two microRNAs (miRs) that decrease AID expression and increase viral infectivity [272]. Thus, antagonism of APOBEC function can occur through virally-encoded proteins or viral non-coding RNAs (Table 1).

In addition, all A3 family genes, except A3A, are significantly upregulated in EBV-positive gastric cancers compared to EBV-negative gastric cancers [273]. Cheng et al. found that BORF2, the ribonucleotide reductase (RNR) large subunit of EBV, interacted with the A3B catalytic domain and inhibited its DNA deaminase activity [274]. Similarly, KSHV encodes the RNR large subunit from the open reading frame 61 (ORF61) protein. Both BORF2 and ORF61 bind to A3B for relocation to perinuclear bodies [274,275]. EBV BORF2 and KSHV ORF61 also interact with A3A, which has a cell-wide distribution, and co-expression of BORF2 or ORF61 leads to A3A relocation into elongated linear structures in the cytoplasm. The major function of RNR is to mediate de novo synthesis of deoxyribonucleotides (dNTPs), an activity commonly encoded by large dsDNA viruses, such as herpesviruses and poxviruses [275,276,277]. Co-immunoprecipitation experiments showed that BORF2 bound to the A3B C-terminal domain. BORF2 inhibited A3B deaminase activity specifically, and cellular levels were stabilized, but the proteasome inhibitor MG132 showed little effect. In gastric adenocarcinoma cells latently infected with EBV, A3B mainly was localized in the pan-nuclear region. However, after EBV reactivation to the lytic cycle, A3B and BORF2 accumulated in nuclear and perinuclear bodies rapidly. These A3B/BORF2 bodies accumulated within the ER. A similar pattern was observed in HeLa, HEK293T, and U2OS cells. Accumulation of C/G-to-T/A mutations (A3B signature) in BORF2-null EBV genomic DNA resulted in lower viral titers and decreased infectivity, suggesting that BORF2 is an important A3B antagonist to maintain EBV genome integrity [274] (Table 1).

Like the gammaherpesviruses, EBV and KSHV, the alphaherpesvirus herpes simplex virus type 1 (HSV-1) encodes a large RNR subunit (ICP6), which binds A3A and A3B and relocalizes these deaminases to the cytosol. Therefore, both alpha- and gammaherpesviruses antagonize A3A and A3B by altering their subcellular localization, suggesting that A3 nuclear functions inhibit herpesvirus DNA replication [274,275]. However, in contrast to results with gammaherpesviruses, deletion of ICP6 had no effect on HSV-1 infectivity. These experiments indicate that HSV-1 may encode another anti-APOBEC activity [275] (Table 1).

Human cytomegalovirus (HCMV), a betaherpesvirus, infects most individuals worldwide. Although HCMV infection in healthy individuals is mostly asymptomatic, HCMV commonly causes congenital infections, which may result in neurological defects, such as deafness or mental retardation in infants [85,278]. HCMV transmission from an infected mother to the fetus occurs through the placenta, resulting in increased A3A expression in infected decidual tissue [78,279]. Weisblum et al. found that A3A levels were highly elevated (~13-fold) by HCMV infection within 24 h by comparisons of ex vivo-infected and uninfected decidual tissues. Induction of A3A was dependent on IFNβ, but not IFNγ treatment. Using the Tet-on system, induced A3A in ARPE-19 epithelial cells significantly decreased HCMV infection, and the HCMV genome was edited by A3A deaminase activity. A3A-mediated HCMV genome editing was detected by 3D-PCR only in the presence of an uracil DNA glycosylase inhibitor (UGI), implying that UNG processes are necessary. HCMV genomes from ex vivo-infected decidual tissue at 7 days post-infection also showed G/C-to-A/T transitions. However, A3A upregulation by HCMV infection was only specific to the decidual tissue. HCMV infection in the chorionic villi within the placenta, primary fibroblasts, and epithelial cells did not induce A3A [78,85]. Such results suggest that A3A is induced in a tissue-specific manner to block congenital HCMV infection.

In support of this idea, HCMV infection has been shown to increase different A3 members in a cell type-specific manner. In HCMV-infected primary human foreskin fibroblasts (HFFs), A3G was induced by IFNβ, yet did not affect HCMV replication. A3G knockout or A3G expression from an adenovirus vector also had no effect [85]. Pautasso et al. showed that A3G hot spots (CCC-GGG) within several HCMV genes were significantly underrepresented, consistent with HCMV selection to limit A3G hot spots and genome deamination [85].

Hepatitis B Virus (HBV) in the *Hepadnaviridae* family is a major cause of liver cirrhosis and hepatocellular carcinoma [280,281]. As a pararetrovirus, HBV has a relaxed circular DNA (rcDNA) genome, which is partially double-stranded and produced by reverse transcription of packaged pregenomic RNA. After infection, rcDNA is repaired to form the covalently closed circular DNA (cccDNA) [282,283]. A3C, A3G, and A3H all are expressed in the liver, a major site of HBV replication [284]. Since A3G was shown to be packaged within HBV cores, Nair et al. confirmed a similar finding with A3A. A3A packaging was not dependent on catalytic activity or DNA-binding ability [283,285]. Overexpression of A3A or A3G did not increase the number of deamination mutations. Interestingly, C-to-T mutational analysis of viral genomes in HBV-producing cells (HepAD38 and HepG2.2.15) showed that the 5′ half of the minus strand was preferentially altered. No mutations were observed in pregenomic RNA, suggesting that deamination occurs after reverse transcription of the minus strand [283] (Table 1).

Type I interferon (IFNα and β) or type III interferon (IFNλ1-3) treatment, which induced A3 enzymes with different kinetics, reduced HBV cccDNA and increased cytidine deamination. Type I and III IFNs increased A3A and A3G, but not A3B levels in hepatocytes [286]. A3B has a higher C-to-T mutational efficiency than A3G within the HBV genome. Multiple heat shock proteins (HSPs), especially Hsp90, stimulate mutational activities of A3B and A3G proteins [287]. Recently, HBV X (HBx) protein has been shown to preferentially decrease cellular A3G protein levels by enhancing their exocytosis. Unlike HIV-1-encoded Vif protein, the effect of HBx on A3G levels is not associated with proteasomal or lysosomal degradation [288].

### 2.2. Deamination-Independent Antagonism of Viruses by APOBEC Enzymes

#### 2.2.1. Deamination-Independent Inhibition of HIV-1

Although studies of Vif-minus HIV-1 first focused attention on A3G-mediated hypermutation, APOBECs also inhibit HIV-1 replication through deamination-independent mechanisms [289,290,291,292]. The widely accepted model for deamination-independent inhibition involves direct binding of A3s to viral genomic RNA to delay primer extension and sterically block the progression of reverse transcriptase (RT). The result is reduced viral cDNA synthesis as well as defects in plus-strand DNA transfer and integration [289,290,291,293,294,295,296,297]. Current models suggest that A3G binds to HIV-1 ssDNA as a monomer to catalyze deamination. Subsequently, A3G forms oligomers through its N-terminal catalytically inactive domains, which then blocks RT elongation of viral ssDNA [298,299]. Particularly striking is the observation that the major mechanism of A3 restriction of HIV-1 infectivity is deamination-independent [300,301,302]. The RT “roadblock” has been proposed to slow down proviral DNA synthesis and increase the time interval available for A3G deamination of ssDNA (Figure 3) (Table 1) [296].

A recent study examined A3G deamination-independent inhibition and the “roadblock” model. These experiments revealed that A3G binds directly to reverse transcriptase, rather than viral RNA, to inhibit HIV-1 DNA synthesis [148,303]. Other experiments showed that various A3 enzymes have different effects on template switching after sterically providing an RT “roadblock” [304]. A3G reduced the template switching of RT, but A3F promoted the template switching of RT due to the high affinity of A3F binding to the primer/template. A3H and A3C did not change the template-switching frequency of RT [304]. During reverse transcription, two separate template-switching events are required to complete synthesis of plus strand DNA [305,306,307]. Either increased or decreased template switching can disrupt the equilibrium of proviral DNA synthesis, thus generating a mutated and non-functional virus independent of deamination [308,309].

Most studies about APOBEC-mediated antagonism of HIV-1 have focused on A3s since these enzymes are expressed in target cells, such as T cells and macrophages. Human A4 did not cause cytidine deamination of HIV-1 cDNA and, in contrast to other APOBECs, even enhanced replication [310]. Nonetheless, a recent study showed that AID, whose role has been largely investigated in class switch recombination and somatic hypermutation in B cells, may be involved in HIV-1 restriction [292]. In transfected or infected 293T cells, AID was shown to bind to the HIV-1 Tat protein, which binds the TAR element within HIV-1 RNA that is essential for viral transcriptional elongation. Subsequently, AID is recruited to RNA exosomes for cleavage of the HIV-1 transcript, potentially inhibiting viral replication [292]. Although AID normally is not expressed in T cells, AID induction by viral infection has been reported [311]. Future studies are still required to investigate whether AID can suppress HIV-1 replication in a physiologically relevant system.

#### 2.2.2. Betaretroviruses and Deamination-Independent APOBEC Activity

Experiments in mice suggested that mA3 has a deamination-independent mechanism for MMTV inhibition [218]. Additional studies showed that mA3 has two isoforms and that both isoforms could be packaged into milk-borne MMTV. BALB/c mice make a longer form of mA3 containing exon 5 compared to mA3 in B6 mice, and there are 15 amino differences between proteins from the two strains. Furthermore, B6 mice express higher levels of mA3 compared to BALB/c mice [182]. Virions isolated from MMTV-infected B6 mammary glands packaged more mA3 than those isolated from BALB/c mice [225]. Although virions produced from mammary cells package mA3, which can inhibit MMTV replication, the primary restriction occurs in lymphoid cells, limiting virus spread to mammary tissue [93]. In vitro deaminase assays showed that the two mA3 isoforms had different substrate preferences. The mA3^B6^ protein showed a higher selectivity at the -2 position for the canonical 5′-TTC-3′ substrate, whereas the mA3^BALB^ enzyme preferred the 5′-ATC-3′ motif with a higher selectivity for the −1 position [225]. Thus, mA3 provides a less restrictive role in BALB/c compared to B6 mice [89,90,91,312].

Our results with the MMTV T-cell-tropic strain, TBLV, also suggest that tumor-derived proviruses from B6 mice have limited differences in their mutation profiles in the presence or absence of Rem compared with those from BALB/c mice [219] (Byun et al., in preparation). Furthermore, TBLV-SD lacking Rem expression shows dramatic acceleration of T-cell tumor induction in B6 mice relative to TBLV-WT, whereas the same experiment in BALB/c mice showed no difference in tumor incidence and latency [219] (Byun et al., in preparation). These results are consistent with mouse strain-specific differences in APOBEC expression and/or activity. Further, our experiments suggest selection for TBLV proviruses with enhanced oncogenic capability in the absence of Rem.

How does mA3 function in vivo during restriction, spread, or selection of MMTV? Although packaged mA3 is catalytically active, only low levels of hypermutation were observed in RIII MMTV virions isolated from the mammary glands of infected BALB/c or B6 mice. Two different studies have suggested how MMTV avoids hypermutation. In the first study, virions isolated from wild-type or mA3-knockout mice were subjected to endogenous reverse transcription reactions. Quantitation of the products revealed that early reverse transcriptase products were most abundant in virions from knockout mice. Furthermore, virions lacking mA3 were more infectious in tissue culture than those from wild-type B6 mice expressing mA3 [225].

In a second study, a different group suggested that MMTV evolved its reverse transcriptase for a high polymerization rate to escape mA3 hypermutation during cDNA synthesis [313] (Table 1). Using cell culture and overexpression experiments, Hagen et al. showed that MMTV packaged the same levels of human A3G or mA3 as HIV-1 lacking Vif expression (HIV-1-ΔVif). However, MMTV accumulated lower levels of G-to-A mutations and was less sensitive to inhibition by A3s than the lentivirus. In a transcomplementation assay, MMTV did not restore the infectivity of HIV-1 ΔVif produced in the presence of either A3G or mA3. They showed in vitro that the MMTV reverse transcriptase (RT) had a higher processivity and a faster rate of DNA synthesis than HIV-1 RT. Using an MMTV RT mutant (F120L), which delayed cDNA elongation, inhibition exerted by A3G and mA3 increased over that observed for wild-type RT. G-to-A mutations also showed a higher frequency with the mutant compared to the wild-type enzyme. These results suggested that the F120L mutation in RT rendered MMTV more susceptible to A3 inhibition, and that this phenotype resulted from elevated G-to-A editing by the mutant. A reduction in the available deoxynucleotide concentration during reverse transcription or direct inhibition with an RT inhibitor predisposed MMTV to mA3 and hA3G restriction. Therefore, the faster polymerization rate of MMTV RT compared to the HIV-1 enzyme was proposed to reduce the availability of MMTV ssDNA and A3-mediated deamination [313].

Hagen et al. [313] also reported that MMTV lacked gene products responsible for antagonizing mA3. Their conclusion was based on a transcomplementation assay in human 293T cells using an HIV-1-ΔVif reporter plasmid in the presence of the pGR102 MMTV proviral clone, which needs confirmation in biologically relevant mouse cells. Moreover, the GR102 clone has been shown to express superantigen, but not other MMTV accessory proteins [314], such as Rem [315,316,317], which antagonizes AID in culture and causes hypermutations typical of mA3 and AID in vivo [219]. Nevertheless, these mechanisms are not mutually exclusive, and MMTV may have evolved multiple mechanisms to counteract different APOBEC enzymes.

The betaretrovirus Mason-Pfizer Monkey Virus (MPMV) has been shown to exclude the host-derived APOBEC protein, rhesus A3G (rA3G), from virion particles [318]. MPMV Gag protein was shown to bind rA3G poorly, which prevented packaging. However, mA3 bound efficiently to MPMV Gag and was packaged into MPMV virions. Therefore, MPMV was sensitive to mA3 and its infectivity was lower in the presence of mA3, leading to the conclusion that exclusion from virions may be a general mechanism used by “simple” retroviruses [318] (Table 1). As noted above, multiple mechanisms are used by retroviruses to avoid APOBEC restriction.

#### 2.2.3. Deamination-Independent Inhibition of Murine Leukemia Viruses

Mouse APOBEC3 restricts infection by exogenous murine gammaretroviruses, such as Friend, Moloney, and AKV murine leukemia viruses (F-MLV, M-MLV, and AKV, respectively) [91,221,319]. With the exception of AKV, most previous studies suggested that the restriction imposed by mA3 on MLV replication primarily occurs through blocks to reverse transcription, not cytidine deamination [320,321,322,323]. Early experiments showed that mutations in mouse APOBEC3 (E73Q and E253Q) abolished catalytic activity and diminished virion packaging in cell cultures [322]. A recent in vivo study used the catalytically inactive mutant (E73Q/E253Q) to generate transgenic mice on the B6 background. The E73Q/E253Q transgene was expressed in M-MLV target cells, including T cells, B cells, pDCs, and macrophages, although at significantly higher levels than endogenous levels. The transgenic E73Q/E253Q mice reduced M-MLV titers similar to that observed in wild-type B6 mice. Viral titers in mA3-knockout mice were significantly increased relative to those in mice carrying a deaminase-inactive mutant [324]. The investigators showed that virions isolated from splenocytes lacked the E73Q/E253Q mutant. They further showed that without virion incorporation, the E73Q/E253Q mutant mA3 in the target cells could restrict infecting viruses [324].

Co-immunoprecipitation of overexpressed M-MLV RT and mA3 in 293T cells revealed that both the wild-type mA3 and the E73Q/E253Q mutant bind reverse transcriptase. Binding was not dependent on RNA since RNase A treatment of the extracts prior to the co-immunoprecipitation did not affect the interaction. Overall, these results confirmed that mA3 restricts retroviral replication independent of deamination activity in vivo [324]. Restriction by mA3 in the target cells prevents incoming viral DNA replication by binding reverse transcriptase rather than binding to viral RNA [324] (Table 1). These results argue that APOBECs restrict retroviruses in both the producer and recipient cells (Figure 3).

How do MLVs counteract deaminase-independent mA3 inhibition? Similar to studies with AKV, the gGag mutant of M-MLV (MLV^gGag^) replicated more poorly in wild-type mice compared to replication observed in mA3-knockout mice. Cores of mutant virus were more unstable than wild-type virus cores, allowing mA3 access to the reverse transcription complex [235] (Figure 3). The MLV^gGag^ mutant virus reverted to yield wild-type gGag by 6 weeks post-infection in B6 mice that express mA3. Reversion did not occur in the mA3-knockout mice, but did occur in transgenic mice on the knockout background expressing the mA3 E73Q/E253Q mutant, which lacks catalytic activity and is not packaged into virions [324]. These results suggest that the primary role of gGag protein is antagonism of mA3 by binding to RT [235,236] (Figure 3) (Table 1). Furthermore, mA3 antagonism by MLV gGag is not dependent on cytidine-deamination activity [324].

A recent study suggested that M-MLV and F-MLV encode a second viral protein that antagonizes APOBEC restriction [325]. Previous experiments suggested that alternative splicing of genomic MLV RNA results in two proteins, p50 and p60, which are translated from the Gag AUG and CUG initiation codons, respectively [326]. The p50 protein, but not the p60 protein, was shown to interact with the C-terminus of mA3. Further, p50-mutant viruses were more restricted than wild-type MLVs in B6 mice compared to mA3-knockout mice. Unlike gGag, which stabilizes MLV capsids, the p50 protein prevents mA3 packaging into virions (Figure 3) (Table 1) [325].

The restrictive role of another APOBEC family member, AID, has been studied using the transforming retrovirus, Abelson murine leukemia virus (Ab-MLV) [327,328]. Gourzi et al. [328] first showed that AID was induced outside germinal centers in response to Ab-MLV infection. In contrast to many APOBEC enzymes, AID is not inducible by type I or II IFNs [327]. AID induction occurred in Ab-MLV-infected bone marrow cells and required reverse transcription of the viral genome. AID induction did not induce hypermutation. Instead, Ab-MLV infection resulted in phosphorylation of the checkpoint kinase-1 (Chk1) to delay cell cycle progression and inhibit proliferation of infected cells. AID also upregulated the NKG2D ligand, Rae-1, on the cell surface, allowing NK cells to eliminate Ab-MLV-infected cells. Therefore, AID did not reduce viral infectivity by mutation or blocking reverse transcription, but restricted the proliferation of cells infected by the transforming Abelson retrovirus [328] (Table 1). Control of infected cell proliferation may be a common mechanism used by APOBECs to limit viral replication (Figure 3).

A subsequent study [327] showed that neither the interferon nor the Toll-like-receptor (TLR) pathways induced AID expression after Ab-MLV infection. Instead, two NF-κB-binding sites within the AID promoter were required for directly recruiting the NF-κB p50 protein to induce transcription during Ab-MLV infection of B cells in the bone marrow [327]. Together, these data suggest that individual APOBEC family members use different mechanisms to activate immune responses after viral infections.

#### 2.2.4. Deamination-Independent Inhibition of Other RNA-Containing Viruses

Enterovirus 71 (EV71) is a single-stranded positive-sense RNA virus (*Picornaviridae* family), which causes hand, food, and mouth disease (HFMD) [329]. In 2018, EV71 was first shown to be inhibited by hA3G [260]. EV71 virus replication in H9 human T cells expressing endogenous hA3G was much lower than in Jurkat T cells that did not express the hA3G protein. In a 293T cell transfection experiment, overexpression of either wild-type A3G or catalytically inactive hA3G inhibited EV71 replication, indicating that restriction is deamination-independent. A3G levels were reduced at 72 h post-infection, suggesting that the virus antagonizes the restriction. A3G inhibition was mediated by competitive binding to the 5′UTR of EV71 RNA with host poly-C-binding protein 1 (PCBP1), a factor that is important for viral RNA synthesis and translation. Other A3 proteins, A3A, A3D, and A3F, but not A3B, A3C, or A3H hapII, inhibited EV71 5′ UTR activity and modestly inhibited EV71 replication. However, EV71 also escaped restriction by inducing the autophagy-lysosomal degradation of A3G. The EV71-encoded 2C helicase protein bound to and induced A3G polyubiquitylation (Table 1). Binding of 2C to A3G occurred through p62/SQSMT1, a ubiquitin-binding adaptor for autophagosome formation, in cytosolic puncta. This study suggests a novel mechanism for antagonism of APOBECs [260].

The activity of A3G against a number of different positive-stranded viruses has been used as the rationale for development of antiviral drugs. The drug IMB-26 previously shown to stabilize A3G levels during HIV-1 infection [330] was the basis for development of a derivative, IMB-Z. IMB-Z inhibits EV71 replication and cytopathic effects [331]. IMB-Z also elevated A3G levels in the late phase of EV71 infection, but did not lead to G-to-A hypermutation of EV71 RNA. The IMB-Z treatment of infected cells also increased endogenous A3G packaging into virions. Unlike the previous study, these investigators suggested that the 3D RNA-dependent RNA polymerase interacted with A3G, rather than other viral proteins, such as the 2C helicase. Although the 3D-A3G interaction was not shown to be critical for antiviral activity [257] (Table 1), A3G is known to bind reverse transcriptase and inhibit retrovirus replication by a non-editing mechanism [324].

The negative-stranded RNA viruses in the *Paramyxoviridae* family (measles, mumps, and respiratory syncytial virus) were inhibited in human A3G-overexpressing monkey cells as well as in T-cell lines expressing A3G at physiological levels [261]. The inhibition of viral infectivity was not dependent on hA3G deamination activity [261]. A subsequent study from the same group [258] showed that in A3G-transduced monkey cells, an endogenous inhibitor of mTORC1 (mammalian target of rapamycin complex-1), REDD1 (regulated in development and DNA damage response-1, also known as DDIT4), was increased. REDD1 reduced measles virus (MV) replication ~10-fold when overexpressed in Vero monkey cells. Rapamycin, which inhibits the mTORC1 pathway, also reduced MV replication to a similar extent as REDD1 overexpression, but a combination did not give an additive effect, suggesting that they function in the same pathway. A3G silencing led to reduced REDD1 expression, and REDD1 silencing in A3G-expressing Vero cells also impaired the A3G-mediated restriction of viral infectivity. IL-2 and phytohemagglutinin (PHA) stimulated hA3G and REDD1 expression in human peripheral blood lymphocytes (PBLs). Silencing of hA3G or REDD1 in stimulated PBLs increased viral titers, confirming the antiviral role of A3G and REDD1. Inhibition of mTORC1 in stimulated PBLs by rapamycin also reduced the viral titers to the level found in non-stimulated lymphocytes [258]. These data suggest that A3G antagonizes some infections by blocking cap-dependent translation of viral mRNAs (Figure 3) (Table 1).

Another negative-stranded RNA virus in the *Orthomyxoviridae* family, influenza A virus (IAV), was previously shown to upregulate A3G, but not A3F, transcription in infected cells. Upregulation was induced by the accumulation of viral RNA in infected cells [332]. Increased A3G transcription was a general IFNβ response to infection that was NF-κB-dependent, but MAP kinase-independent. However, even with strong induction, A3G did not appear to inhibit IAV replication [332]. A more recent study reached a similar conclusion that A3 proteins likely did not induce IAV hypermutation to restrict infection [333]. Therefore, IAVs probably are not affected by APOBEC proteins, but knockdown or knockout of *APOBEC* genes may be informative for their effects on IAV replication.

Very few studies have investigated the effects of the APOBEC family members, A2 and A4. However, the A4 cytidine deaminase was shown to inhibit the paramyxovirus, Newcastle Disease Virus (NDV). A4 affected NDV replication in chickens by reducing viral RNA levels [98] (Table 1). As mentioned previously, overexpression of A4 actually enhanced HIV-1 replication, primarily at the level of transcription [310]. An intriguing possibility is that AID and A3 enzymes evolved primarily to restrict retroelements, whereas other cytidine deaminases diversified to control replication of other pathogens.

#### 2.2.5. Deaminase-Independent Antagonism of DNA-Containing Viruses

In vivo experiments using mA3-knockout mice with transgenic hA3 alleles suggested that A3s do not affect HSV-1 replication or genome deamination [334]. The failure to detect deamination may have resulted from the use of HSV-1 strains that encoded effective A3 inhibitors (see previous sections). In contrast, infection by a single-stranded DNA virus, minute virus of mice (MVM), was inhibited by transgenic hA3A, but not hA3G. This result is consistent with the nuclear replication of MVM and the presence of hA3A in the nucleus. MVM inhibition by hA3A was independent of deamination as detected by the 3D-PCR method [334] (Table 1). Previous experiments indicated that replication of the parvovirus adeno-associated virus (AAV) was inhibited by hA3A, but not a catalytic site mutant of hA3A [335]. A recent bioinformatic study revealed that 22% of all human viral genomes exhibit a footprint of A3 enzymes. In particular, the B19 erthyroparvovirus showed a dramatic A3 footprint [336]. Therefore, some parvoviruses appear to be restricted using an A3 deaminase-dependent mechanism, whereas others are deaminase-independent.

Most studies have reported that the HBV genome is edited by A3 proteins. However, HBV replication also appears to be inhibited by a deaminase-independent mechanism [337,338,339,340,341,342]. Results in tissue culture cell lines suggest that RNA-DNA hybrid synthesis is impaired by A3G. A block to early minus-strand formation was observed. Further, catalytically inactive A3G inhibited HBV DNA synthesis [338], suggesting a mechanism similar to that observed with retroviruses (Table 1).

Polyomaviruses are small, non-enveloped DNA viruses that cause or are associated with various human diseases, including nephropathy and Merkel cell carcinoma [343]. Infection by different polyomaviruses, including BKPyV, JCPyV or MCPyV, induces A3B, indicating that increased A3B levels are a general consequence of viral infection. Polyomavirus large T antigen is sufficient for A3B induction by regulation of Rb and E2F [343,344]. However, A3B induction does not affect BKPyV replication and infection, possibly due to reduced numbers of the A3B-preferred motifs, TCA/TGA trinucleotides, in the viral genome [343].

## 3. APOBEC Proteins and Restriction of Endogenous Viruses and Retrotransposons

APOBECs restrict several types of DNA elements within mammalian genomes that spread through reverse transcription. These elements include long terminal repeat (LTR)-containing retrotransposons, also known as endogenous retroviruses (ERVs), and non-LTR retrotransposons, such as long interspersed element 1 (LINE-1 or L1) or Alu [345,346,347,348,349,350,351,352,353,354,355]. These endogenous retroelements proliferate within their host genomes by a copy-and-paste mechanism that involves reverse transcription, which exposes ssDNA to APOBEC activity. Similar to their effects on retroviruses, APOBECs cause G-to-A positive strand editing of ERVs and retrotransposons as well as physical interference with retrotransposition. The net effect is reduced expression and mobilization of these elements (reviewed in [350,351,355]). Although the unrestricted movement of endogenous viruses and transposons leads to widespread mutagenesis, it has been estimated that ~100 human genes partially consist of retrotransposons [356]. Thus, retrotransposition provides both detrimental and advantageous consequences for their hosts.

### 3.1. ERVs and APOBEC-Mediated Inhibition

ERVs compose approximately 8 to 10% of the human and mouse genomes [349,357,358]. Due to the accumulation of mutations, ERVs, such as HERV-Ks, have become mostly inactive in humans [355,359]. Reconstructions of intact HERV-K genomes and assays in cell culture revealed that these ancient retroviruses accumulated many C/G to T/A mutations due to the activity of A3 proteins [360]. A recent study measured the activity of opossum A1 expressed in HeLa cells using reporter assays. Expression of this A1 reduced the mobilization of the LTR retrotransposon MusD in a deamination-dependent manner. Mutation of the A1 catalytic site abolished this effect [361]. Using two strains of mice, Toll-like receptor 7 (*Tlr7*) knockout mice, which show emergence of ERVs as well as *hA3G*^+/+^
*mA3*^−/−^ mice [362], the effect of transgenic hA3G was assessed on ERVs. By comparison of transgenic *hA3G*^+/+^
*mA3*^−/−^
*Tlr7*^−/−^ mice on the B6 background to control mice lacking hA3G expression, hA3G, but not mA3, restricted ERV emergence [362]. In contrast, the MusD LTR retrotransposon was not suppressed by human A3G or mA3 expressed in *Tlr7*^−/−^ mice [362]. However, AID expression in HeLa cells reduced MusD retrotransposition [348]. Therefore, both in vitro and in vivo experiments demonstrate that individual APOBECs restrict different LTR retrotransposons, but the relative contribution of various APOBECs for restricting LTR retrotransposons in humans needs further investigation.

Recent bioinformatic analyses have revealed enrichment of G-to-A editing in ERVs within genomes from different species [363,364]. The strand specificity of APOBEC editing, which causes G-to-A mutations of retroviral sense strands, also has been observed in ERVs across vertebrate genomes. G-to-A mutations are 10-fold greater than C-to-T transitions, a result expected from A3 targeting [363]. Enrichment of these edited elements was observed in genic regions that were preferentially exonized, but the edited elements were not enriched in any specific biological pathways or gene ontology term, suggesting that editing is a general mutational mechanism. LTRs regulate gene expression, and edited LTR retrotransposons are enriched within transcription start sites and promoter regions [363]. These data suggested that APOBEC-induced hypermutations of LTRs provide a selection to reduce threats to genome function [363].

Another study of 160 mammalian species showed that ERVs drive the evolution of A3 genes [364]. A strong positive correlation was observed between A3 Z-domain copy numbers and the extent to which G-to-A mutations have accumulated in ERV sequences. Mammalian species that have accumulated more ERVs tend to have higher A3 Z-copy numbers, and the A3 subfamily amplification occurred concurrently with prominent ERV invasions in primates [364]. These bioinformatic analyses suggest coevolution of APOBEC family enzymes with ERVs.

### 3.2. Non-LTR Retrotransposons and APOBECs

LINE-1/L1 elements are retrotransposons that lack envelope genes and LTRs, but they move through reverse transcription and are subject to APOBEC-mediated antagonism. Non-LTR retrotransposons comprise approximately 30% of the human genome [349]. Most analyses of APOBEC effects on non-LTR retrotransposons have focused on L1 and the related short interspersed elements (SINEs), e.g., Alu. Several members of the APOBEC family (AID, A1, A3A, A3B, A3C, A3D, A3G, A3F, and A3H) have been shown to inhibit both L1 and Alu retrotransposition using deaminase-independent as well as deaminase-dependent mechanisms [346,348,352,365,366,367,368,369,370,371,372,373,374,375].

AID and its catalytically inactive mutant both have been shown to inhibit L1 retrotransposition, indicating that restriction does not occur through deamination and hypermutation [348]. The DNA binding and nuclear localization functions of AID were not required for restriction of L1 retrotransposition, suggesting that reverse transcription of these elements and APOBEC inhibition occur in the cytosol [348].

All A3 family proteins have been demonstrated to inhibit L1 and/or Alu retrotransposition, [352,366,367,368,369,370,371,372,373,374,376], yet different levels were observed with various deaminases using an overexpression assay in 293T cells [372]. A3G restriction of Alu retrotransposition was less than that observed for A3A and A3B, but higher than A3H-mediated restriction [372]. Except for A3A, other A3 proteins restrict these non-LTR retrotransposons in a deamination-independent manner [369,371,372,373,374,376]. By using an UNG inhibitor, which inhibits the UNG-dependent DNA repair-mediated degradation pathway, Richardson et al. [370] showed that A3A inhibition of L1 involved deamination of cytidines in the L1 single-stranded (-) DNA during reverse transcription. A deaminase-defective A3A mutant could not inhibit L1 retrotransposition [335,346,352,367,368], suggesting that A3A suppresses L1 activity through a deamination-dependent mechanism.

A3B is a nuclear cytidine deaminase expressed in human embryonic stem cells (hESCs) and induced pluripotent stem cells (iPSCs) [335,352,368]. A3B knockdown in HeLa and hESC cell lines resulted in a strong increase in L1 retrotransposition. However, knocking down the expression of other APOBEC3 proteins (A3C, A3D, A3F, and A3G) failed to affect L1 retrotransposition in hESCs [377]. Deaminase oligomerization and RNA binding were required for A3C inhibition of L1 retrotransposition [371]. Co-immunoprecipitation experiments showed an RNA-dependent physical interaction between L1 ORF1p, which is equivalent to the Gag protein of retroviruses. A3C dimers were essential for L1 restriction, and overexpression of A3C reduced L1 reverse transcription activity [371]. A3D (also known as A3DE) interacted with L1 ORF1p in the cytosol to target L1 ribonucleoprotein particles in an RNA-dependent manner [373]. A3D inhibited L1 reverse transcriptase activity to a greater extent than A3B [373]. The N-terminal 30 amino acids of A3G were required for oligomerization and to inhibit L1 and Alu retrotransposition [372]. However, based on experiments using *hA3G*^+/+^
*mA3*^−/−^
*Tlr7*^−/−^ B6 transgenic mice (discussed above), human A3G does not suppress L1 in vivo [362], consistent with the failure to suppress L1 retrotransposition by A3G knockdown [377]. Earlier studies also indicated that A3G has low or no anti-L1 activity [367,368,369,378].

A3H was shown to have weak anti-L1 activity [372,376], but the A3H variants (G105R, K121E, and E178D), which have the highest incidence among sub-Saharan Africans, demonstrated strong anti-L1 activity [376]. A3A, A3B, A3D, and A3H all restricted Alu retrotransposition [367,376]. A3A and A3H reduced Alu retrotransposition through a mechanism that did not involve the sequestration of Alu RNA within high-molecular-mass complexes [346,376]. However, most studies of different A3 proteins have been performed in cell culture using APOBEC overexpression and reporter assays. Experiments using physiological conditions to test the relative contributions of A3 proteins to L1 and Alu retrotransposition are needed.

Recent genomic analysis has provided more information about the roles of APOBECs in restricting L1 in vivo [379,380]. Bioinformatics of human cancer DNA revealed a negative correlation between the expression levels of A3 members (A3C/D/F/H) and L1 insertions, suggesting that these proteins may effectively restrict L1 retrotransposition [380]. Additional analysis showed that A3A/B hotspot 5′-ATC-3′ motifs may have evolved from their capability for damaging L1 while minimizing host gene damage [379].

In a retrotransposition assay based in cell culture, A1 was shown to reduce the mobility of L1 by a deaminase-independent mechanism [375]. The A1 enzyme was shown to bind L1 RNA, and there was an association of A1 with the L1 RNP complexes [375]. A more recent study using opossum A1 revealed that this marsupial A1 inhibited the human L1 retrotransposition independent of its catalytic deamination activity [361]. Therefore, the majority of APOBEC enzymes appear to restrict non-LTR retrotransposons in the absence of hypermutations.

## 4. Conclusions

Viruses have provided many insights into host cell functions, and the APOBEC family proteins are no exception. Cytidine deaminases have evolved to serve multiple purposes, including host-specific functions, such as lipid metabolism, as well as defense against infectious viruses and retrotransposons. Evolutionary studies of the APOBEC cytidine deaminases have revealed the early appearance of A2 and AID, yet only AID has demonstrable deaminase activity. Interestingly, virus-mediated antagonism of APOBECs can be divided into deaminase-dependent and deaminase-independent mechanisms, even with respect to the same viral target and APOBEC family member. In fact, deaminase-independent mechanisms appear to have been the first to evolve to inhibit retrotransposons. Although the deamination-independent mechanisms appear to be prevalent, they have not been as widely studied as the mechanisms of deamination. The mechanistic choice may depend on host and/or cell-specific factors. APOBEC members often are inducible, particularly through interferons, which produce an antiviral response within the infected cells to provide an early warning system for their uninfected neighbors. Nevertheless, APOBECs have dual effects. Just as AID generates hypermutation of the immunoglobulin variable region for antibody affinity maturation and antiviral activity, APOBEC enzymes also generate viral diversification. RNA editing by A1 or other APOBECs with different cofactors could induce tissue-specific antiviral activities. From RNA-containing coronaviruses and retroviruses to DNA-containing herpesviruses and HBV, APOBECs provide a means for control of virus replication, but also harbor the potential for emergence of new pathogens and viral resistance to drugs or host immune responses.

## Figures and Tables

**Figure 1 microorganisms-08-01899-f001:**
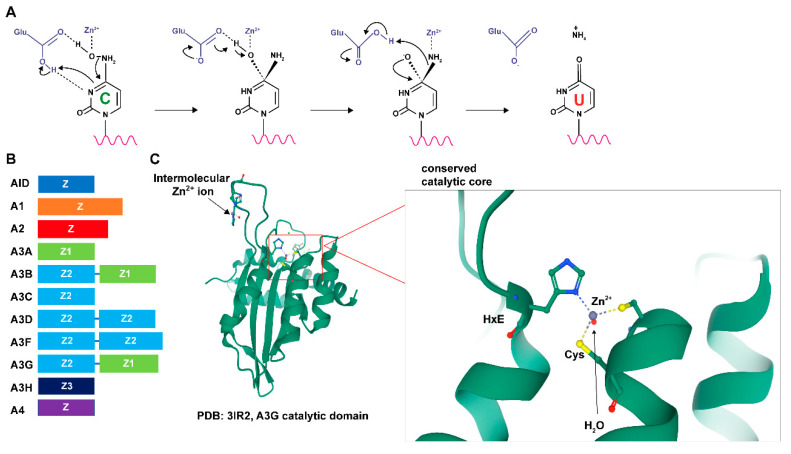
Structural characteristics and activity of APOBECs. (**A**) Reaction of APOBEC-mediated cytidine deamination (see text for details). (**B**) Different APOBECs are composed of distinct zinc-coordinating (Z) domains. In the mammalian A3 family, Z domains are categorized into three distinct phylogenetic groups, Z1, Z2, and Z3. Members of the human APOBEC genes and the relative lengths of their Z domains are shown. (**C**) A ribbon diagram of the A3G catalytic domain shows its globular structure and an expanded view of the zinc-coordinating active site (from PDB 3IR2) [10]. Zinc ions are depicted in gray. The expanded view shows the interactions of the histidine ring and cysteine side chains with zinc. The red dot near the zinc ion represents a water molecule.

**Figure 2 microorganisms-08-01899-f002:**
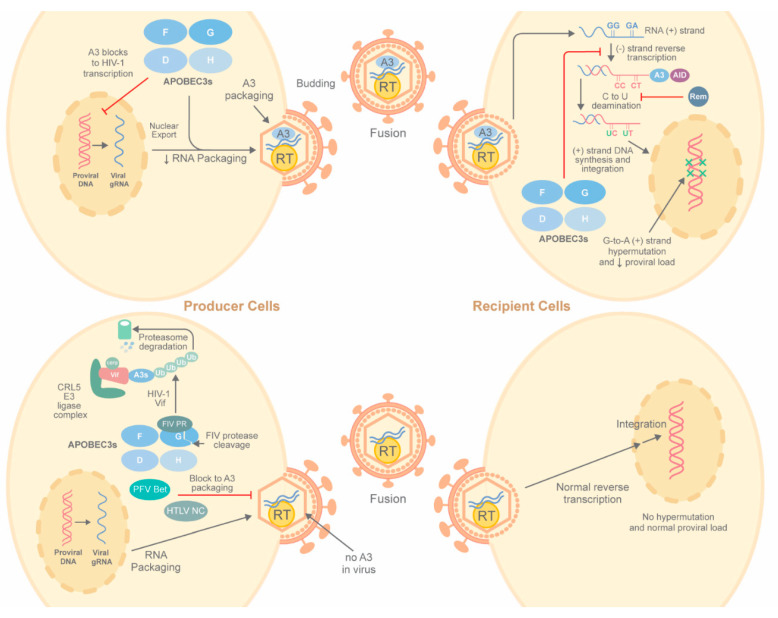
Deamination-dependent restriction of viruses by APOBECs. In the absence of an antagonist, such as HIV-1 Vif, virus-producing cells (**top left**) often package A3 proteins, including A3D, A3F, A3G and A3H, (abbreviated as D, F, G, and H) into virions together with reverse transcriptase and two copies of genomic RNA. A3 enzymes (specifically A3G) also may block HIV-1 transcription, lowering the amount of viral RNA available for packaging. Subsequent infection of susceptible recipient cells (**upper right**) leads to reverse transcription of viral plus-strand RNA. During minus-strand DNA synthesis, C-to-U deamination occurs, leading to G-to-A hypermutation on the viral DNA plus strands. Blocks to reverse transcription also occur, resulting in integrated viral DNA with increased G-to-A transition mutations and reduced proviral loads. Virus-producing cells (**lower left**) that make an antagonist, such as Prototype Foamy Virus (PFV) Bet or Human T-Cell Leukemia Virus type 1 (HTLV-1) nucleocapsid (NC), fail to package A3. Other A3 antagonists, such as HIV-1 Vif, recruit an E3 ligase complex to A3 enzymes to mediate their ubiquitylation and degradation, whereas the Feline Immunodeficiency Virus (FIV) protease (PR) reportedly cleaves A3 proteins. In either case, A3 levels are reduced sufficiently to prevent virion incorporation. A3-free virions then mediate successful reverse transcription without hypermutation in recipient cells (**lower right**). Mouse Mammary Tumor Virus regulator of export of MMTV mRNA (MMTV Rem) expression leads to AID proteasomal degradation (not shown), but AID is not incorporated into virions in the presence or absence of Rem. RT = reverse transcriptase; gRNA = genomic RNA; CRL = cullin–RING ubiquitin ligase.

**Figure 3 microorganisms-08-01899-f003:**
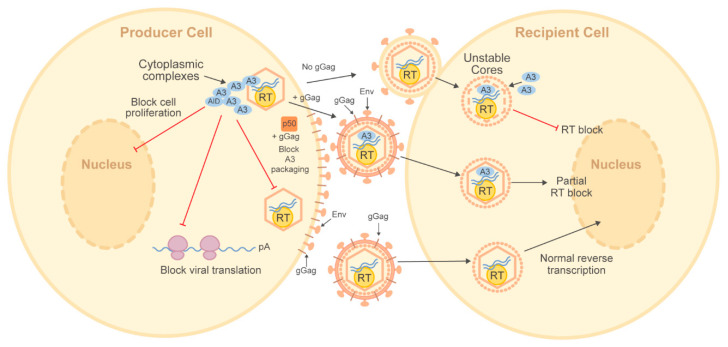
Deamination-independent inhibition of viral replication. In virus-producing cells, A3 and AID proteins are present in cytoplasmic complexes (**left**). AID is a shuttling protein and its nuclear activity may block cell proliferation and the number of infected cells. Alternatively, multiple APOBEC proteins may inhibit viral RNA translation. In the case of murine leukemia viruses, several different virally specified inhibitors, such as p50 and glycoGag (gGag) are produced in infected cells. The p50 protein prevents A3 incorporation into virions, whereas gGag is incorporated into virus particles to promote their stability, allowing normal proviral DNA synthesis in recipient cells (**right**). If gGag is present in viral particles, A3 may be incorporated into virions, leading to reduced proviral DNA. If virion RNA is packaged in the absence of gGag, viral cores are unstable and allow A3 enzymes in recipient cells to block reverse transcription. RT = reverse transcriptase.

**Table 1 microorganisms-08-01899-t001:** APOBEC Proteins and their Viral Inhibitors ^1^.

Viral Family	Virus	APOBEC Protein	Mechanism of APOBEC Action	Viral APOBEC Antagonist	Mechanism of Viral Antagonist
*Coronaviridae*	SARS CoV-1/CoV-2	hA1, hA3	Hypermutation of viral RNA??	Nucleocapsid??	??
*Hepadnaviridae*	HBV	hA3	Hypermutation; reduced viral DNA	HBx	Decreased hA3 levels by enhancing exocytosis
*Herpesviridae*	EBV	hA3	Hypermutation; reduced viral DNA	BORF2	Inhibited deaminase activity; altered A3 subcellular localization
*Herpesviridae*	HCMV	hA3	Hypermutation; reduced viral DNA	??	??
*Herpesviridae*	HSV-1	hA3	Reduced viral DNA?	ICP6??	Altered hA3 subcellular localization
*Herpesviridae*	KSHV	hA3	Hypermutation; reduced viral DNA	ORF61	Altered hA3 subcellular localization
*Herpesviridae*	KSHV	hAID ^2^	Induced NKG2D ligands on NK cells	Viral miRs	Decreased hAID protein
*Paramyxoviridae*	MV	hA3	Blocked translation of viral mRNA	??	??
*Paramyxoviridae*	NDV	chA4 ^2^	Reduced viral RNA	??	??
*Parvoviridae*	MVM	hA3	Reduced viral DNA	??	??
*Picornaviridae*	EV71	hA3	Inhibited viral RNA synthesis and translation	2C helicase; 3D RNA-dependent RNA polymerase	Autophagy-lysosomal degradation of A3G by 2C; reduced virion packaging by 3D??
*Retroviridae*	Ab-MLV	mAID	Inhibited proliferation of infected cells; induced NKG2D ligands on NK cells	??	??
*Retroviridae*	AKV	mA3	Hypermutation; reduced viral DNA	gGag	??
*Retroviridae*	FFV	fA3	Hypermutation; reduced viral DNA	Bet	Sequestered fA3 reduces virion packaging
*Retroviridae*	FIV	fA3 ^2^	Hypermutation; reduced viral DNA	Vif; PR	fA3 degradation and cleavage
*Retroviridae*	HIV-1	hAID	Inhibition of viral transcription	??	??
*Retroviridae*	HIV-1 SIV	hA3/ sA3 ^2,3^	Hypermutation, reduced transcription, increased defective proviruses, reduced viral DNA synthesis by binding RT; mutated viral epitopes for CTLs	Vif; Env; Vpr	hA3 degradation and reduced virion packaging; decreased A3 translation; increased Pol packaging
*Retroviridae*	HTLV-1	hA3	Hypermutation; inactivation of Tax	Nucleocapsid	Reduced hA3 packaging into virions; oligoclonal expansion of infected cells
*Retroviridae*	M-MLV	mA3	Reduced viral DNA by binding RT	gGag	Unstable cores
*Retroviridae*	M-MLV, F-MLV	mA3	Reduced viral DNA	p50	Reduced mA3 packaging into virions
*Retroviridae*	MMTV	mAID ^2^	Hypermutation; reduced proviral DNA	Rem	mAID degradation
*Retroviridae*	MMTV	mA3	Reduced viral DNA	RT	Rapid polymerization
*Retroviridae*	MMTV	mA3	Hypermutation??	Rem	??
*Retroviridae*	MPMV	rA3 ^2^	??	??	Reduced rA3 packaging into virions

^1^ See text for references and details. ^2^ h = Human; m = murine; s = simian; f = feline; r = rhesus; ch = chicken. ^3^ Various human and primate A3 proteins have different inhibitory activities and susceptibility to Vif-mediated degradation. A3 proteins may act synergistically. ‘??’ means unknown or unclear.

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
