# Peer review of "The Role of APOBECs in Viral Replication"

_microorganisms, 2020, doi:10.3390/microorganisms8121899_

Round 1
Reviewer 1 Report
This is a nicely written and organized review of the Apobec gene family. It is, however, very dense, with only 3 figures and no tables to summarize the very detailed text. It might be useful, for example, to add a table listing the AID/APOBEC genes and their restriction activity against the various viruses as covered in the text, and/or to list the various viral antagonists. The experimental details showing that APOBECs do not have any effect on influenza could be eliminated or reduced to a single sentence. There is also some inconsistency in verb tense usage in some paragraphs that impacts flow and clarity.
Other comments:
1 - Legends for Figures 2 and 3 use undefined abbreviations or names that are eventually defined in the text, but only many pages later (for example PFV and p50). Should be defined in the legend.
2 – Line 67: I see no reference to C6666 in the cited reference.
3 - Lines 71-73 :“In vitro, A1 binds RNAs with a preference for AU-rich sequences, whereas much greater deamination specificity is observed in tissues. This observation has been explained by the discovery that A1 deaminase activity in vivo requires an RNA-binding component.”
It is not clear how deamination specificity can be greater than the preference for AU rich sequences in RNA. There should also be citations for these statements.
4 - Lines 114 – 116, 121 – 122: “In in vitro experiments, A3G binding to RNA oligonucleotides displaced A3G binding to DNA and inhibited deaminase activity”
“RNAs of at least 25 nucleotides in length inhibit both A3G binding to ssDNA and deaminase activity [36]”
These sentences say the same thing, it would be more informative to the readers if they were combined.
5 - Line 160 states that “For dual domain-containing APOBECs such as A3G, the N-terminal domain is catalytically inactive”, but Line 234 states that “mA3 has the catalytic domain within the N-terminus”
6 - Lines 163-164: “Mutations of amino acid residues around the pseudo-catalytic site of the A3G N-terminal domain display reduced deamination rates compared with the wild-type A3G”
This sentence reads as “mutations of amino acid residues display reduced deamination rates” It should be corrected for clarity.
7 - Lines 176 – 177: “CD1 also may participate in nucleic acid binding and dimerization/oligomerization”
In the paragraph that ends with the statement above, the authors talk about multiple studies that show that CD1 is involved in both dimerization and nucleic acid binding. This conclusion sentence seems to downplay those results without giving the readers any reason (e.g. more studies are needed).
8 - Lines 182- 183: “Overexpression of A4 in yeast and bacteria did not yield cytidine deaminase activity on DNA”
A transition sentence/phrase would help readers in switching the discussion from A2 to A4.
9 - Lines 220 – 222: “IFNɣ, a type II IFN that is primarily expressed by NK cells and activated T cells [77], increases A3G expression in monocytes and macrophages [78,79] as well as A3A in the presence of TNFα [78,80].”
Last part of this sentence is disconnected from the verb.
10 - Line 262: “…or is shuttled back to the”
Missing “AID” before “is”.
11 - Line 286: “resulted in in G-to-A mutations” should only be one “in”.
12 - Figure 2: Top left scheme indicates that APOBEC3s block HIV-1 transcription, but this was only shown for A3G. Could be clarified in legend.
13 - Line 397: “A3G-targeted motif (5’-CCC-3’) was mutated up to 6-fold”
Underlining should not extend to 3’.
It’s not clear what the compared condition is when the authors say “mutated up to 6 fold”.
14 - Line 513: “Higher Pol levels yielded faster virus DNA replication to protect the virus genome from A3G-mediated hypermutation”
HIV-1 reverse transcription is not DNA replication.
15 - Lines 529-531: “Unlike HIV-1, HTLV-1 has an accessory protein, the HTLV-1 basic leucine zipper factor (HBZ), which is encoded on the proviral minus strand and is required for the maintenance of HTLV-1-transformed cells.”
This sentence makes it sound like HIV-1 doesn’t have any accessory proteins
16 - Lines 533-534: “Although HTLV-1 hypermutation is rare in vivo, a small percentage of genomes (0.1-0.5%) are edited extensively, and up to 97% of cytidines are deaminated”
It should make it clear that 97% of cytidines in the extensively edited genomes are deaminated
17 - Line 729: RNR of herpesviruses stands for ribonucleotide reductase. This was indicated in Line 746 and redefined as RNR
18 - Line 745: HSV-1 stands for Herpes Simplex Virus 1
19 - Line 757: “Weisblum et al. found that A3A was highly elevated (~13-fold) by HCMV within 24 hours”
“A3A levels” and “HCMV infection” would be clearer.
20 - Lines 934 and 953: These lines begin new paragraphs and therefore are making distinctive points and yet refer to “ this group” and “from the same group”. Should name the groups, rephrase or combine with previous paragraph.
21 - The authors variously refer to the C57BL/6 strain as B6 or C57BL/6, but should uniformly use B6 after the first usage.
Reviewer 2 Report
This is an excellent and comprehensive review on APOBECs system in viral replication. Two minor suggestions can be considered to improve the review.
- In abstract, 'Such viral antagonists often are only partially successful, leading 23 to selection for viral variants.' This sentence is unclear and hard to be understood.
- Section 2.1.6: Some new evidences have indicated that SARS-CoV [1] and SARS-CoV-2 [2,3] also undergo APOBECs editing, I would suggest to include these evidences in the review.
1.Wang, S. M., & Wang, C. T. (2009). APOBEC3G cytidine deaminase association with coronavirus nucleocapsid protein. Virology, 388(1), 112-120.
2.Di Giorgio, S., Martignano, F., Torcia, M. G., Mattiuz, G., & Conticello, S. G. (2020). Evidence for host-dependent RNA editing in the transcriptome of SARS-CoV-2. Science Advances, eabb5813.
3.Wang, R., Hozumi, Y., Zheng, Y. H., Yin, C., & Wei, G. W. (2020). Host immune response driving SARS-CoV-2 evolution. Viruses, 12(10), 1095.
